# Small intestinal flora graft alters fecal flora, stool, cytokines and mood status in healthy mice

Yinyin Xie[1], Linyang Song[2], Junhua Yang[2,3] , Taoqi Tao[1], Jing Yu[4], Jingrong Shi[5], Xiaobao Jin[3]

**Fecal microbiota transplantation is widely used. Large intestinal microbiota (LIM) is more similar to fecal microbiota than small intestinal microbiota (SIM). The SIM communities are very different from those of LIM. Therefore, SIM transplantation (SIMT) and LIM transplantation (LIMT) might exert different influences. Here, healthy adult male C57Bl/6 mice received intragastric SIMT, LIMT, or sterile PBS administration. Microbiota graft samples were collected from small/ large intestine of healthy mice of the same age, sex, and strain background. Compared with PBS treatment, SIMT increased pellet number, stool wet weight, and stool water percentage; induced a fecal microbiota profile shift toward the microbial composition of the SIM graft; induced a systemic anti-inflammatory cytokines profile; and ameliorated depressive-like behaviors in recipients. LIMT, however, induced merely a slight alteration in fecal microbial composition and no significant influence on the other aspects. In sum, SIMT, rather than LIMT, affected defecation features, fecal microbial composition, cytokines profile, and depressive-like behaviors in healthy mice. This study reveals the different effects of SIMT and LIMT, providing an interesting clue for further researches involving gut microbial composition change.**

## Introduction

The adult mammalian gastrointestinal tract harbors numerous and complex microorganisms, the gut microbiota, which creates an enormous and dynamic ecosystem in its host. The gut microbiota is well known as an integral part of its host (Bäckhed et al, 2005) for its essential role in regulating lots of aspects of the host during homeostasis (Jandhyala et al, 2015) and disturbance (Pickard et al, 2017). Nowadays, most studies have focused on the effects of the gut microbiota on intra-gastrointestinal change (Rubbens et al, 2020), nutrient absorption and metabolism (Dabke et al, 2019), systemic immune status (Hand et al, 2016), endocrine status (Neuman

et al, 2015), the development of the brain (Cowan et al, 2019), and behavior (Rincel et al, 2019).

Recently, fecal microbiota transplantation has been well established in lots of basic studies and clinical treatments of many disease conditions, such as *Clostridioides difficile* infections (Drekonja et al, 2015), inflammatory bowel disease (Paramsothy et al, 2017), cirrhosis (Bajaj et al, 2019), metabolic syndrome (de Groot et al, 2017), and depression-related disorders (Cai et al, 2019). This supports that fecal microbiota transplantation is an effective way to better uncover the influence of this microbiota on their host.

To the best of our knowledge, the gut microbiota transplanted in the experiments or clinical treatments reported by the related scientific publications were derived from feces (Tang et al, 2017). Compared with microbiota in the small intestine, microbiota in the large intestine is more similar to fecal microbiota (Gu et al, 2013; Zhao et al, 2015). Therefore, fecal microbiota transplantation may be esteemed to some extent to explore the effects of the microbiota dwelling in the large intestine. In several animal and human studies (Gu et al, 2013; Scheithauer et al, 2016; Yuan et al, 2020), notably, they all reported that the communities of small intestinal microbiota (SIM) were different from those of large intestinal microbiota. For example, in mice, the facultative anaerobic bacteria such as *Lactobacillus* were more enriched in the small intestine, while strictly anaerobic bacteria including *Lachnospiraceae*, were more enriched in the large intestine (Gu et al, 2013). This prompted us to hypothesize that SIM and large intestinal microbiota transplantation (LIMT) may exert different influences on their host.

In this study, healthy recipient mice received the small intestinal microbiota transplantation (SIMT) or the LIMT. Then we observed some common physiological, immune, endocrine, and behavioral indicators (including body weight, food and water intake, defecation features, the composition of fecal microbiota, cytokines, the stress hormone, and depressive-like behaviors) to investigate whether different effects exist between the SIM and the LIMT. This study is the first to compare the effects of SIMT and LIMT on the host, which contributes to dissect the relationship between gut microbiota and host from a new perspective unfocused before.

[1]Class 3, Grade 2018, School of Clinical Medicine, Guangdong Pharmaceutical University, Higher Education Mega Center, Guangzhou City, People's Republic of China [2]Department of Anatomy, School of Biosciences and Biopharmaceutics, Guangdong Pharmaceutical University, Higher Education Mega Center, Guangzhou City, People's Republic of China [3]Guangdong Key Laboratory of Pharmaceutical Bioactive Substances, Guangdong Pharmaceutical University, Higher Education Mega Center, Guangzhou City, People's Republic of China [4]Editorial Department of Journal of Sun Yat-sen University, Guangzhou City, People's Republic of China [5]Department of Data Mining and Analysis, Guangzhou Tianpeng Technology Co., Ltd, Guangzhou, PR China

Correspondence: jhyang2018@gdpu.edu.cn

# Results

### No significant differences between three groups of recipient mice in health status before microbiota transplantation

To evaluate the health status of the recipient and donor mice and to see whether they were similar in health status indexes including body weight, food intake, water consumption and defecation features. On postnatal day (PND) 75, the data of body weight, food intake, and water consumption were collected as described in the section with heading "Measurement of food intake, water intake, and body weight alteration." One-way ANOVA revealed no significant differences among the control (CON) group, SIMT group, and LIMT group before microbiota transplantation (indicated by the names Pre-CON group, Pre-SIMT group, and Pre-LIMT group) in food intake ($F_{2,15}$ = 0.054; n = 6; $P$ = 0.947), water intake ($F_{2,15}$ = 0.076; n = 6; $P$ = 0.927), and body weight ($F_{2,15}$ = 0.127; n = 6; $P$ = 0.882) (Fig 1A–C).

On PND76, the data of defecation features were collected as described in the section with heading "Analyses of number of stool pellets, stool wet weight, and stool water content." One-way ANOVA revealed no significant differences before microbiota transplantation among the Pre-CON, Pre-SIMT, and Pre-LIMT in stool pellets number ($F_{2,15}$ = 0.026; n = 6; $P$ = 0.974), stool wet weight ($F_{2,15}$ = 0.078; n = 6; $P$ = 0.925), and stool water percentage ($F_{2,15}$ = 0.002; n = 6; $P$ = 0.998) (Fig 1D–F).

### No significant differences between recipient mice and donor mice in health status before microbiota transplantation

To investigate whether there was potential significant differences between the donor mice and the recipient mice before microbiota transplantation in the six health status indexes as described in the section with heading "No significant differences between three groups of recipient mice in health status before microbiota transplantation," we first merged the data of the Pre-CON, Pre-SIMT, and Pre-LIMT groups (n = 6) into one group (n = 18) here and then compared it with the data of the donor mice (n = 20) using $t$ test. The results showed no significant differences (all $P$-values > 0.05) (Fig 2A–F).

### Neither SIMT nor LIMT affected food intake, water consumption, and body weight alteration

On PND92, the data of food intake, water consumption, and body weight alteration were collected as described in the section with heading "Measurement of food intake, water intake and body weight alteration." One-way ANOVA revealed no significant alterations after microbiota transplantation between the CON, SIMT, and LIMT groups in food intake ($F_{2,15}$ = 0.033; n = 6; $P$ = 0.968), water intake ($F_{2,15}$ = 0.149; n = 6; $P$ = 0.863), and body weight alteration ($F_{2,15}$ = 0.474; n = 6; $P$ = 0.631) (Fig 3A–C).

### Only SIMT significantly affected defecation features of recipient mice

Significant differences were found between three groups after microbiota transplantation in stool pellets number ($F_{2,15}$ = 4.089; n = 6; $P$ = 0.038), stool wet weight ($F_{2,15}$ = 6.09; n = 6; $P$ = 0.012), and stool water percentage ($F_{2,15}$ = 16.604; n = 6; $P$ < 0.001) (Fig 3D–F). Least significant difference (LSD) post hoc analyses revealed that the SIMT group had a larger number of pellets (SIMT versus CON: 28 ± 4.36 versus 19.33 ± 4.82, $P$ < 0.05), a larger stool wet weight (SIMT versus CON: 0.83 ± 0.14 versus 0.55 ± 0.11, $P$ < 0.05) and a larger stool

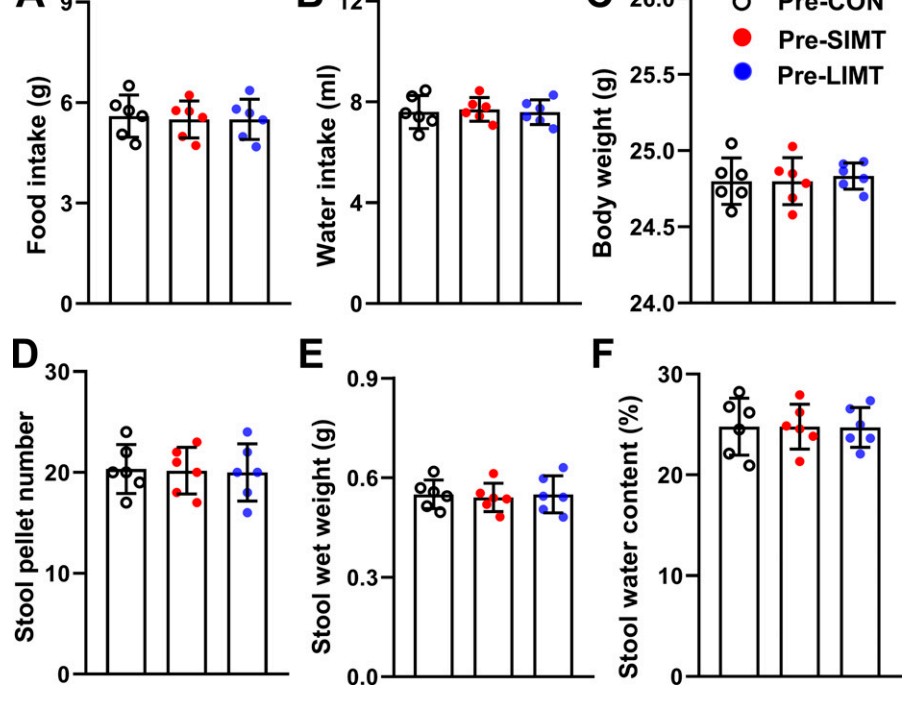

**Figure 1. No significant differences between three groups of recipient mice in health status before microbiota transplantation.**
**(A)** Bars represent average value of food intake amount of each group. **(B)** Bars represent average value of water intake amount of each group. **(C)** Bars represent average value of body weight of each group. **(D, E, F)** Bars represent average value of stool pallet number (D), stool wet weight (E), and stool water content (F) of each group. All data were analyzed using one-way ANOVA followed by least significant difference (LSD) post hoc test. n = 6. Data were shown in figures as mean ± SD.

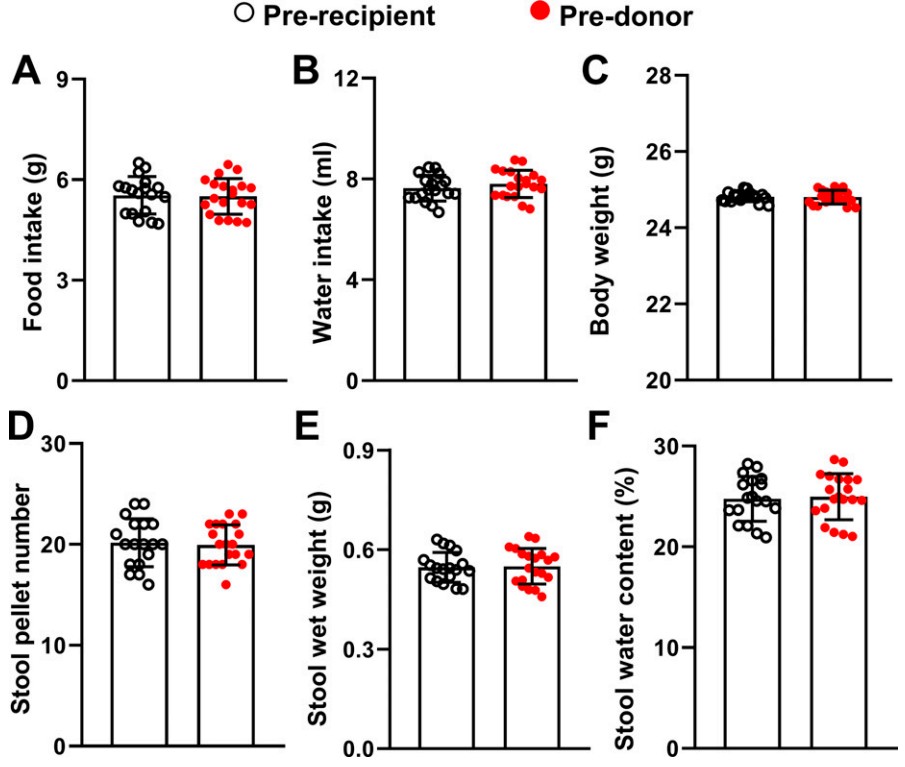

○ **Pre-recipient**    ● **Pre-donor**

Figure 2.    No significant differences between recipient mice and donor mice in health status before microbiota transplantation.
**(A)** Bars represent average value of food intake amount of each group. **(B)** Bars represent average value of water intake amount of each group. **(C)** Bars represent average value of body weight of each group. **(D, E, F)** Bars represent average value of stool pallet number (D), stool wet weight (E), and stool water content (F) of each group. All data were analyzed using *t* test. n = 20 for donor mice group; n = 18 for recipient mice group. Data were shown in figures as mean ± SD.

water percentage (SIMT versus CON: 38.85 ± 4.39 versus 25.03 ± 5.65, *P* < 0.001) (Fig 3D–F). LSD post hoc analyses revealed no significant differences between the LIMT and CON groups in all three indexes (all *P*-values > 0.05) (Fig 3D–F).

### No significant differences in microbial community composition between three groups of recipient mice before microbiota transplantation

Cluster analysis was used to identify groups of fecal samples that contained similar microbiota composition and to compare the similarity or heterogeneity between samples from three groups of recipient mice before microbiota transplantation. It generated a dendrogram, without any cluster consisting of single group–derived samples or primarily one group–derived samples (Fig 4). The results of ANOSIM analysis of the 18 recipient samples showed no significantly different microbial community composition in different groups (*P* = 0.918). These findings suggest no significant differences between three groups of recipient mice before microbiota transplantation.

### Only SIMT induced a significant shifting of fecal microbiota profile compared with that before transplantation

As shown in Fig 5A, the results of principal co-ordinates analysis (PCoA) analysis of the fecal samples from the CON and SIMT recipient mice (collected both before and after microbiota transplantation) showed a significantly different microbial community composition before and after microbiota transplantation (*P* =

0.019). In Fig 5A, there was an absence of overlap between the scatters distribution of the fecal samples collected before (pre-SIMT, shown in purple scatters) and after (SIMT, shown in red scatters) microbiota transplantation in SIMT recipient mice. On the contrary, the results of PCoA analysis of the fecal samples from the CON and LIMT recipient mice (collected both before and after microbiota transplantation) showed no significant difference in microbial community composition before and after microbiota transplantation (*P* = 0.181) (Fig 5B).

### SIMT induced fecal microbiota profile shift towards the microbial composition of the small intestinal microbiota graft (SIMG)

Cluster analysis was used to identify groups of fecal samples that contained similar microbiota composition and to compare the similarity or heterogeneity between samples from the recipients and graft samples. It generated a dendrogram, grouping the 18 recipients' samples and the two graft samples into two distinct clusters at operational taxonomic unit (OTU) levels (Fig 6). Cluster 1 included the large intestinal microbiota graft sample (LIMG) and all fecal samples from the CON and LIMT groups. Cluster 2 included the intestinal microbiota graft sample (SIMG) and all fecal samples from the SIMT group. To determine whether SIMT/LIMT altered the overall composition of the fecal microbial community at the OTU level, a two-dimensional heat map of the total of 832 OTUs is shown in Fig 6 to further illustrate the distinct patterns of fecal microbial composition in the 18 recipients' samples and the two graft samples.

The results of PCoA analysis with ANOSIM analysis based on Bray–Curtis distance of the 18 recipient samples showed a significantly

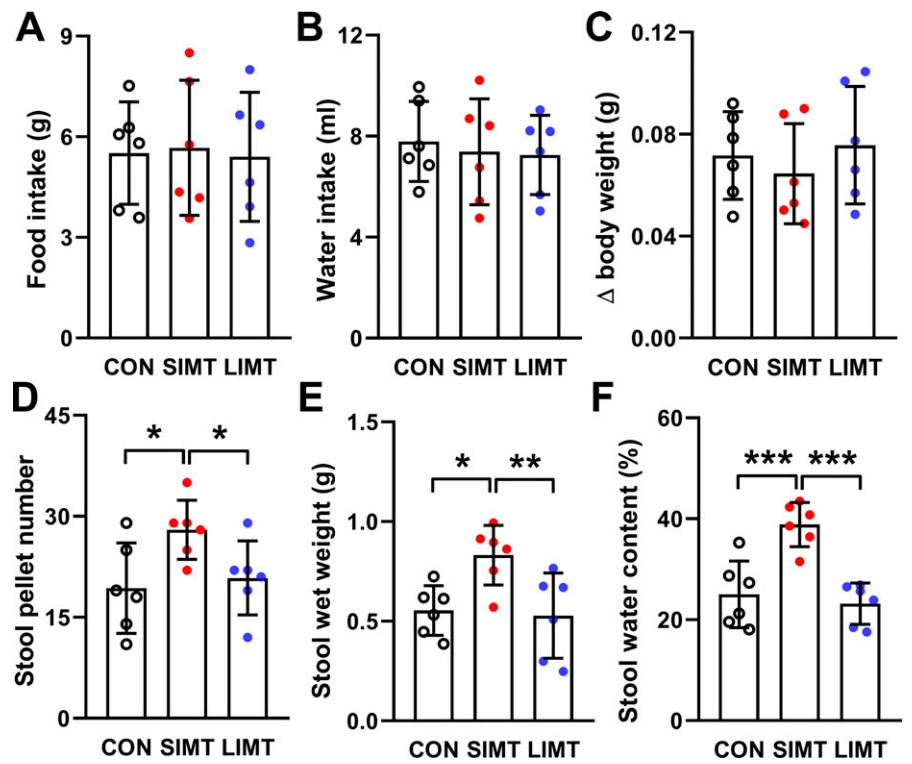

**Figure 3. Small intestinal microbiota transplantation under physiological conditions affected defecation features of mice.**
**(A)** The seven raw measured values of food intake amount of each animal were first averaged and thus an individual average food intake amount value (IAFIA) was produced. All IAFIAs were shown in as the scatter points. Then the mean of the six IAFIAs in each group were calculated and the mean was present as the bars. **(A, B)** The data of water intake amount were calculated and presented here using the same procedure as that used in (A). **(C)** Bars represent average value of the extent of body weight change in each of three groups. **(D, E, F)** Bars represent average value of stool pallet number (D), stool wet weight (E) and stool water content (F) in each of three groups. All data were analyzed using one-way ANOVA followed by LSD post hoc test. *$P < 0.05$; **$P < 0.01$; ***$P < 0.001$. n = 6. Data were shown in figures as mean ± SD.

different microbial community composition in different groups ($P$ = 0.001) (Fig 7). The results of the community bar plot analysis of samples from 18 recipients and samples from two grafts demonstrated that the SIMT group samples, than the CON group or LIMT group samples, had a more similar microbial community composition to that of the SIMG (Fig 8). These findings shown in Figs 6–8 suggest consistently that SIMT under physiological conditions induced fecal microbiota profile shift towards the microbial composition of the SIMG.

To further determine whether there was a statistically significant difference in fecal microbial composition among the three recipient groups, the Kruskal–Wallis $H$ test and Bonferroni post hoc test analysis were performed for each of the 832 OTUs (Table S1). The results revealed 211 OTUs that showed a significant difference among the three recipient groups ($P < 0.05$) (Table S2). Of the 211 OTUs, 29 OTUs were more abundant in the SIMG (Table S2). On the contrary, 98 OTUs were more abundant in the LIMG (Table S2). The rest 84 OTUs were detectable neither in the SIMG nor in the LIMG (Table S2).

A total of 47 OTUs showed a significant difference between the SIMT and CON groups, with 10 ones increased and 37 ones decreased in the SIMT group (Table S3). Interestingly, all the 10 increased OTUs were also more abundant in the SIMG and the 37 decreased OTUs were also more abundant in the LIMG (Table S3). These findings confirm that SIMT in physiological conditions induced fecal microbiota profile shift towards the microbial composition of the SIMG.

Unlike in SIMT, merely six OTUs showed a significant difference between the LIMT and CON groups, with four ones increased and two ones decreased in the LIMT group (Table S4). Specifically, three of the four increased OTUs were more abundant in the LIMG, whereas the rest one was more abundant in the SIMG (Table S4). Moreover, one of the two decreased OTUs were more abundant in the LIMG, whereas the rest one was more abundant in the SIMG (Table S4). These results showed that LIMT did not result in a dramatic or consistent shift of overall fecal microbiota composition toward that of the LIMG.

### SIMT induced an anti-inflammatory cytokine profile in blood

Significant differences were found among three groups in the levels of serum interferon (IFN)-γ ($F_{2,15}$ = 3.822; n = 6; $P$ = 0.046), IL-1β ($F_{2,15}$ = 4.715; n = 6; $P$ = 0.026), TNF-α ($F_{2,15}$ = 6.343; n = 6; $P$ = 0.010), and IL-4 ($F_{2,15}$ = 6.311; n = 6; $P$ = 0.010), and no significant differences were found in that of IL-6 ($F_{2,15}$ = 0.283; n = 6; $P$ = 0.758) and IL-10 ($F_{2,15}$ = 2.023; n = 6; $P$ = 0.167) (Fig 9A–F). LSD post hoc analyses revealed that the SIMT group had fewer IFN-γ, IL-1β, TNF-α, and more IL-4 in serum than the CON group ($P$-values < 0.05 for IFN-γ and IL-1β; $P$-values < 0.01 for TNF-α and IL-4) (Fig 9A–F). LSD post hoc analyses revealed no significant differences between the LIMT and CON groups in all cytokines (all $P$-values > 0.05) (Fig 9A–F).

### SIMT resulted in no significant alterations in corticosterone level in the blood

There were no significant alterations among three groups in the levels of serum corticosterone ($F_{2,15}$ = 1.572; n = 6; $P$ = 0.240) (Fig 9G), suggesting that neither SIMT nor LIMT under physiological conditions influenced hypothalamic–pituitary–adrenal (HPA) axis activation.

### SIMT decreased depressive-like behaviors

Three groups of mice were subjected to sucrose preference test (SPT), forced swimming test (FST), and tail suspension test

# Community heatmap analysis on OTU level

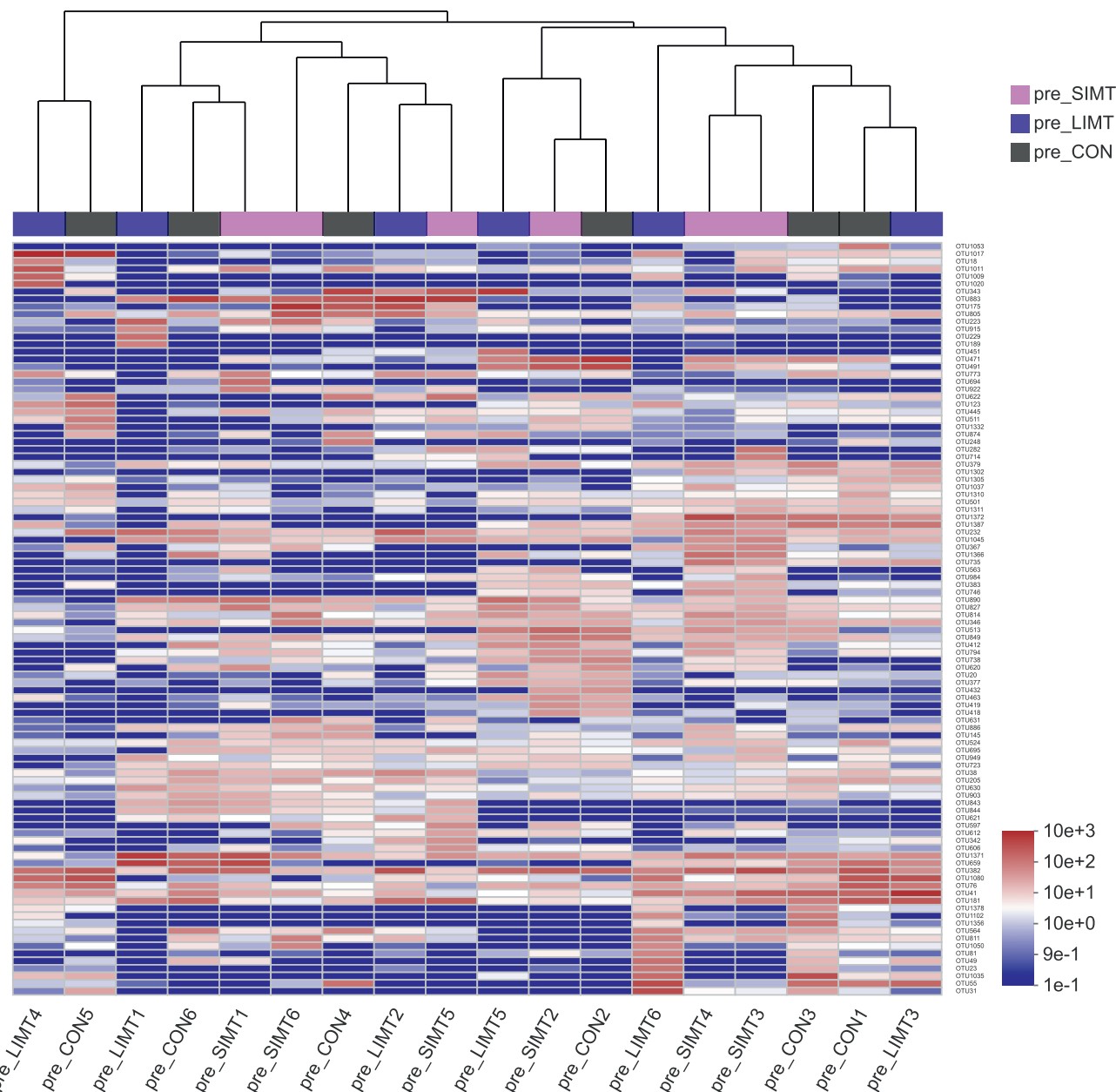

**Figure 4. Heat map and hierarchical clustering analysis of the fecal samples collected from recipient mice before transplantation.**
Hierarchical clustering based on Bray–Curtis distance matrix is shown at the top of the heat map made of the data from the 18 fecal samples obtained from three groups before transplantation. The identity document (ID) information of animal was provided at the bottom of the heat map. The ID information of each operational taxonomic unit (OTU) was provided on the right of the heat map that are visible when magnified. pre_CON, mice assigned to control group measured before microbiota transplantation; pre-SIMT, mice assigned to small intestinal microbiota transplantation group measured before microbiota transplantation; pre-LIMT, mice assigned to large intestinal microbiota transplantation group measured before microbiota transplantation.

(TST). Each of these tests showed a significance in Kruskal–Wallis $H$ test (SPT: $H$ = 20.979; $df$ = 2; $P < 0.001$) (Fig 10A) or one-way ANOVA (FST: $F_{2,33}$ = 9.276; n = 12; $P < 0.001$; TST: $F_{2,33}$ = 12.889; n = 12; $P < 0.001$) (Fig 10B and C). Post hoc analyses revealed that the SIMT group had more sugar consumption (Bonferroni; $P < 0.001$) (Fig 10A) and

had less immobility time in FST and TST (LSD; FST: $P < 0.01$; TST: $P < 0.001$) (Fig 10B and C) compared with the CON group. Post hoc analyses revealed no significant differences between the LIMT and CON groups in all three indexes (all $P$-values > 0.05) (Fig 10A–C).

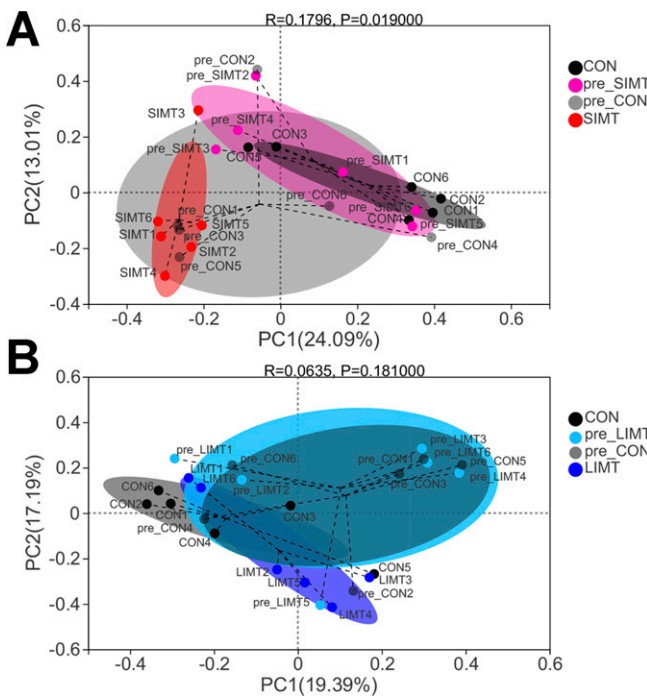

**Figure 5. Two-dimensional principal co-ordinates analysis (PCoA) of Bray–Curtis distance matrix revealed a significant difference in fecal microbiota composition between pre_small intestinal microbiota transplantation (SIMT) and SIMT fecal samples.**
**(A)** Two-dimensional PCoA of Bray–Curtis distance matrix was conducted for the data from pre_CON, pre_SIMT, CON, and SIMT fecal samples. **(B)** Two-dimensional PCoA of Bray–Curtis distance matrix was conducted for the data from pre_CON, pre_large intestinal microbiota transplantation, CON, and large intestinal microbiota transplantation fecal samples.

## Discussion

It was found in this study that SIMT led to significant alterations in defecation features, fecal microbial composition, cytokine profile, and depressive-like behaviors, whereas LIMT induced merely a slight alteration in fecal microbial composition and failed to influence all the other aspects. Hence, the present study provided evidence for our hypothesis that SIM and large intestinal microbiota may exert different influences on a series of physiological aspects in their host.

Given the well-known effects of gut microbiota on stool characteristics (Huang et al, 2018), immune status (Hand et al, 2016), and mood-related behaviors (Zheng et al, 2016), the shift of fecal microbial composition may explain the alterations in defecation features, cytokines profile, and depressive-like behaviors in SIMT mice. Furthermore, we deduced that the microbial being more abundant in the SIMG (indicated by higher OTU percentage than in the LIMG) play an important role in the shift of fecal microbial composition of recipients toward the SIMG.

Gut microbiota transplantation has been demonstrated to have significant effects on obesity, anorexia and stress-related diseases (de Clercq et al, 2019; Lee et al, 2019; Zhang et al, 2019). The data of this study, however, showed that gut microbiota transplantation, whether SIMT or LIMT, brought no change in body weight, food intake, water intake, or blood corticosterone levels. Accordingly, we

speculate that the effect of gut microbiota transplantation on body weight, metabolism, and HPA axis activation is not obvious under physiological conditions.

Although analysis at the level of the genus was frequently used to reflect the functional change of gut microbiota (Arumugam et al, 2011; Feng et al, 2019; Sims et al, 2019), analysis at the OTU level could provide the most detailed and precise information for the estimation of the shift in fecal microbial composition. Therefore, we performed an analysis of fecal microbial composition and graft microbial composition at the OUT level.

In this study, all the 10 increased OTUs in the SIMT group than in the CON group were also more abundant in the SIMG. Of the 10 increased OTUs in the SIMT group, three OTUs (OTU634, OTU542, and OTU768 in our data) belong to the genus *Lactobacillus*. The genus *Lactobacillus*, verified to improve the motility of the intestine (Wang et al, 2019) and exert an anti-inflammatory effect (Forsberg et al, 2014; Devi et al, 2018; Pan et al, 2018; Mata Forsberg et al, 2019), is widely considered to be beneficial for stress response and depression (Yong et al, 2019). Moreover, probiotics, including the genus *Lactobacillus*, have been widely regarded to benefit lots of physiological and pathological aspects (Lomasney et al, 2014; Kuhn et al, 2020). The role of the genus *Lactobacillus* reported previously might be one of the reasons for the effects induced by SIMT in our study. Other gut microbiota, such as the *Enterococcus faecalis* (OUT804 in our data), were previously linked to decreased anxiety-like and depressive-like behaviors in mice (Takahashi et al, 2019; Kambe et al, 2020). Besides, the possibility should not be ruled out that the strains that did not show differences among groups also had an impact on the observed indicators because the interaction between intestinal microbiota and the body is undoubtedly extremely complex (Arumugam et al, 2011), involving a series of physiological and biochemical mechanisms and processes. In 2011, the concept of enterotypes appeared in a report (Arumugam et al, 2011), which said that the enterotypes are mostly determined by species composition, but major molecular functions are not necessarily provided by species of high abundance. The concept of enterotypes highlighted the importance of analyzing the function of microbial communities as a whole. Therefore, we inclined to the view in the current study that the intestinal microbiota in the graft interacted with the host as a whole, rather than by one or several specific microbiota contained in the graft. This study aimed to investigate whether there are different effects of SIMT and LIMT on a series of physiological indexes and the results addressed this issue sufficiently. More researches are needed to further explore exactly how many strains of microbiota are involved in mediating the observed effects and how the precise interactions between the strains in the graft and those in the host are carried out.

In our study, the ratio of the quantity of SIM to that of large intestinal microbiota used in transplantation was approximately equal to the physiological ratio in a healthy mouse organism, rather than 1:1. If the ratio was adjusted to 1:1, the required quantity of large intestinal microbiota used for transplantation would be reduced to the same level of SIM because the small intestine harbors much less microbiota than the large intestine in normal conditions (Kastl et al, 2020). Hence, LIMT might also induce no significant effects even in the case of the ratio being 1:1.

## Community heatmap analysis on OTU level

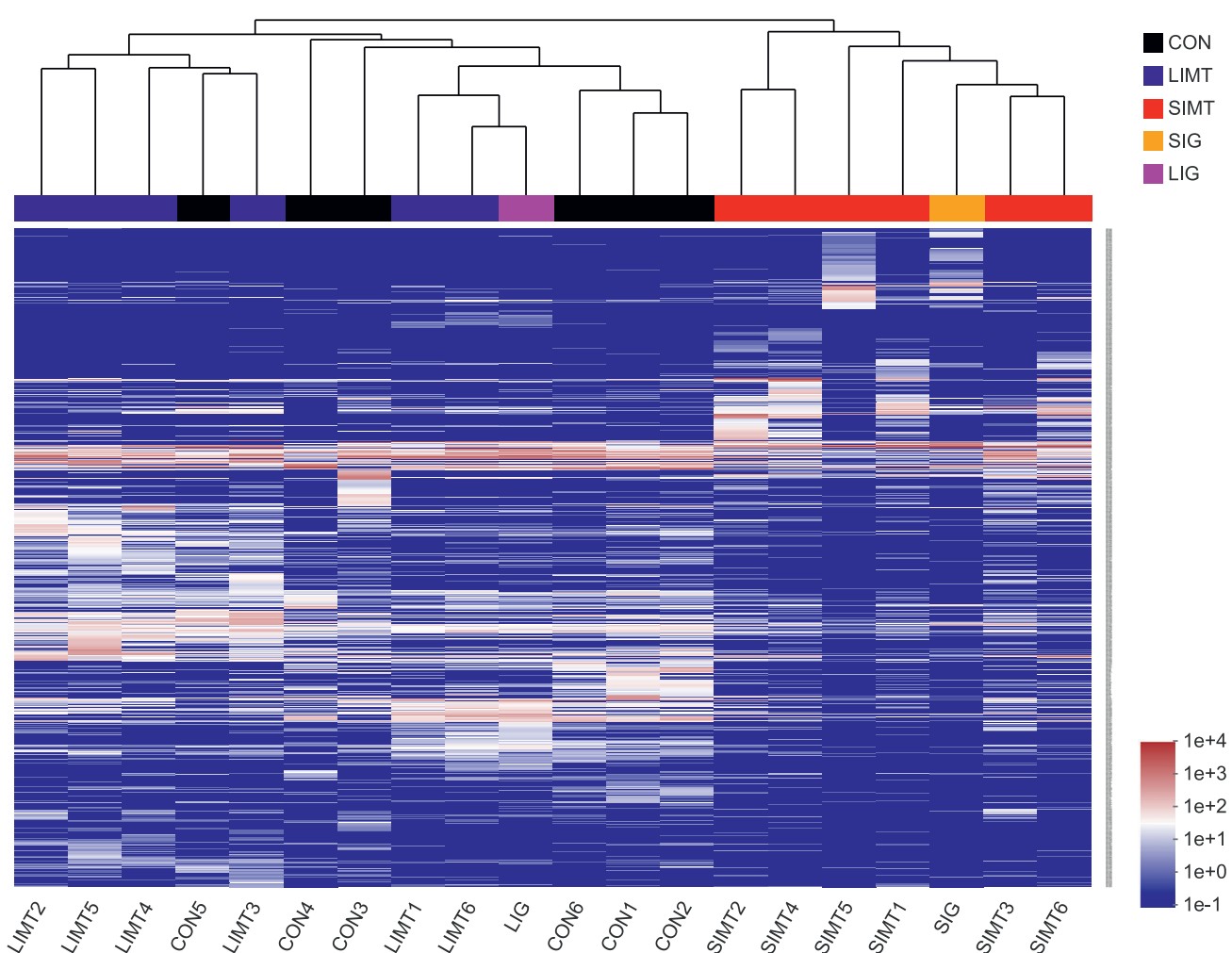

**Figure 6.  Heat map and hierarchical clustering analysis of the fecal samples and graft samples.**
Heat map of the all 832 OTUs in the 18 fecal samples from three groups and two graft samples. Hierarchical clustering based on Bray–Curtis distance matrix is shown at the top of the heat map. The ID information of animal was provided at the bottom of the heat map. The ID information of each OTU was provided on the right of the heat map that are visible when magnified. CON, CON group; LIMT, LIMT group; SIMT, SIMT group; SIMG, the small intestinal microbiota graft sample; LIMG, the large intestinal microbiota graft sample.

Most of the studies concerning gut microbiota (Org et al, 2016), immunity (Taneja, 2018) and mood status (Kokras & Dalla, 2014) took gender factors into account. However, only male mice were used in this study. Moreover, all mice were housed individually to avoid their oral communication of gut microbiota with one another through coprophage, which occur through sharing cage and bedding when housed together (Kenagy & Hoyt, 1979). Both singly housing and repeated intragastric administration procedure caused slight stress. Female C57Bl/6 mice are more susceptive to stress-induced neurobehavioral alterations, such as dysregulation of the HPA axis, whereas male C57Bl/6 mice were reported to be of more resilience (Hodes et al, 2015; Marchette et al, 2018; Palumbo et al, 2020). Given that the current study aimed at comparing the effects of SIMT and LIMT in healthy host or under physiological condition, only male C57Bl/6 mice thus were used in our study. Sexual dimorphism exists throughout the whole animal kingdom (Deng & Jasper, 2016) and sexual dimorphism in gut microbiome is described in the literature and thought to be mainly driven by sex hormones (Cui et al, 2019; Ma & Li, 2019). Therefore, the conclusion of the present study may not be applicable simply to female mice, although there are possibly similar effects by SIMT or LIMT in female mice.

In this study, intragastric transplantation was used, in which the graft first entered the small intestine before entering the large intestine with downstream of the contents in the small intestine. The environment in the small intestine may be more suitable for bacterial colonization of small intestine microbiota graft. In contrast, bacteria in large intestinal microbiota graft were likely not easy to survive or thrive in the small intestine and therefore failed to exert significant effects. This may explain the significant effects were only observed in the small intestine microbiota transplantation group. Given the most wide utility of intragastric transplantation in clinical practice and animal experimental research in gut/fecal microbiota transplantation, using intragastric transplantation allows best us to compare the results of our study with

**Figure 7. Two-dimensional PCoA of Bray−Curtis distance matrix revealed significant difference in fecal microbiota composition among groups.**

previous reports, and ensures best that our findings could be referred to in the future basial and even clinical researches in this field.

Other factors might also underlie the different effects induced by small and large intestine microbiota transplantation. For instance, the difference in the composition of small intestine bacterial graft and large intestine bacterial graft lies mainly in the relative abundance of different bacteria because most high-abundance bacteria exist both in the small intestine and large intestine merely with different relative abundance values. Therefore, the bacteria, at least the common bacteria, in both grafts will share the opportunity to survive and reproduce in the small intestinal

environment. In this line, so is in the large intestinal environment. What's more, the interaction between the grafted and host-holding microorganisms, the interaction between these microorganisms and intestinal mucosa, immune system, nutrition, or metabolism are all very complex processes hard to be clearly explored. These processes are very likely to participate in the different effects induced by small and large intestine microbiota transplantation in our study. In conclusion, it is indeed difficult to reveal exactly the mechanism underlying the different effects induced by small and large intestine microbiota transplantation. Nevertheless, the data obtained in our study are sufficient to support our scientific hypothesis that small intestine microbiota transplantation and large intestine microbiota transplantation may exert different influences on their host.

This study contributes to understanding the relationship between gut microbiota and host from a new perspective unfocused before. Despite the difficulty to carry out SIMT in the clinic, the conclusion of this study provides an interesting clue for animal experimental researches involving dysbacteriosis. Also, this study suggests that the intake of probiotics or a certain designed diet may drive the gut microbiota profile to shift towards the profile of small intestine microbiota, which may bring a beneficial effect, especially in case of constipation, inflammatory diseases, and mood disorders.

# Materials and Methods

### Animals and study design

Male specific pathogen−free C57BL/6 mice at the age of PND56 were ordered from Guangdong Medical Laboratory Animal Center (Foshan, China). The animals were housed in the specific pathogen−free room

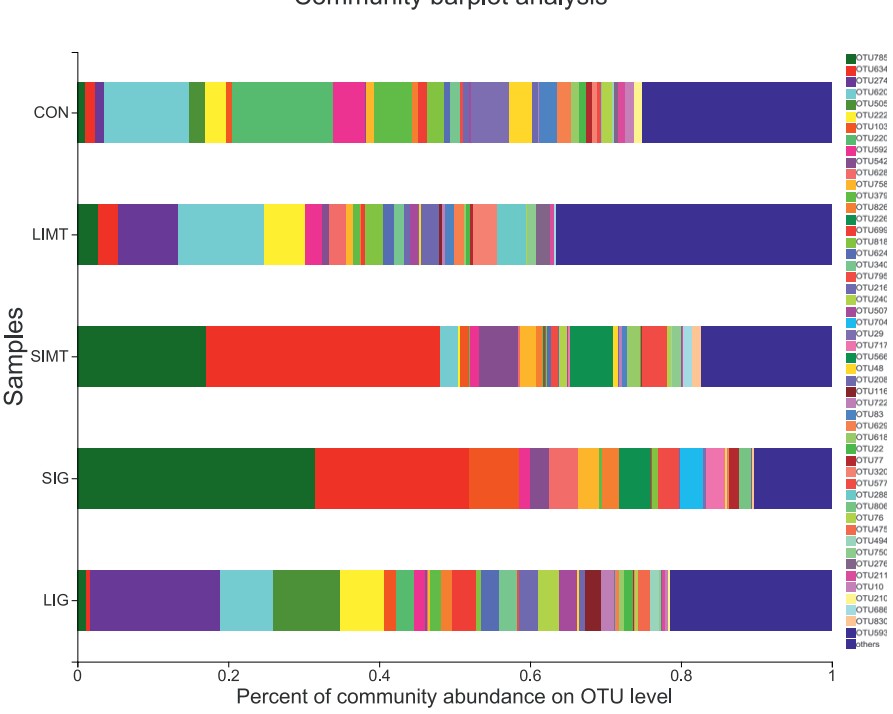

**Figure 8. The microbiota composition in the fecal samples and the graft samples at the OUT level.**
Each OTU was assigned to a certain color as shown on the right of the figure. The percentage of each OTU shown in the CON, small intestinal microbiota transplantation, and large intestinal microbiota transplantation groups represents the median of the six raw individual percentage data in each of the three groups. The percentage of each OTU shown in the small intestinal microbiota graft and LIMG samples represents the raw percentage data. The relative abundance of 51 OTUs presented were defined as those that were more than 1% of total sequences across all samples combined; the remainder were merged and lumped in a category designated as "Other."

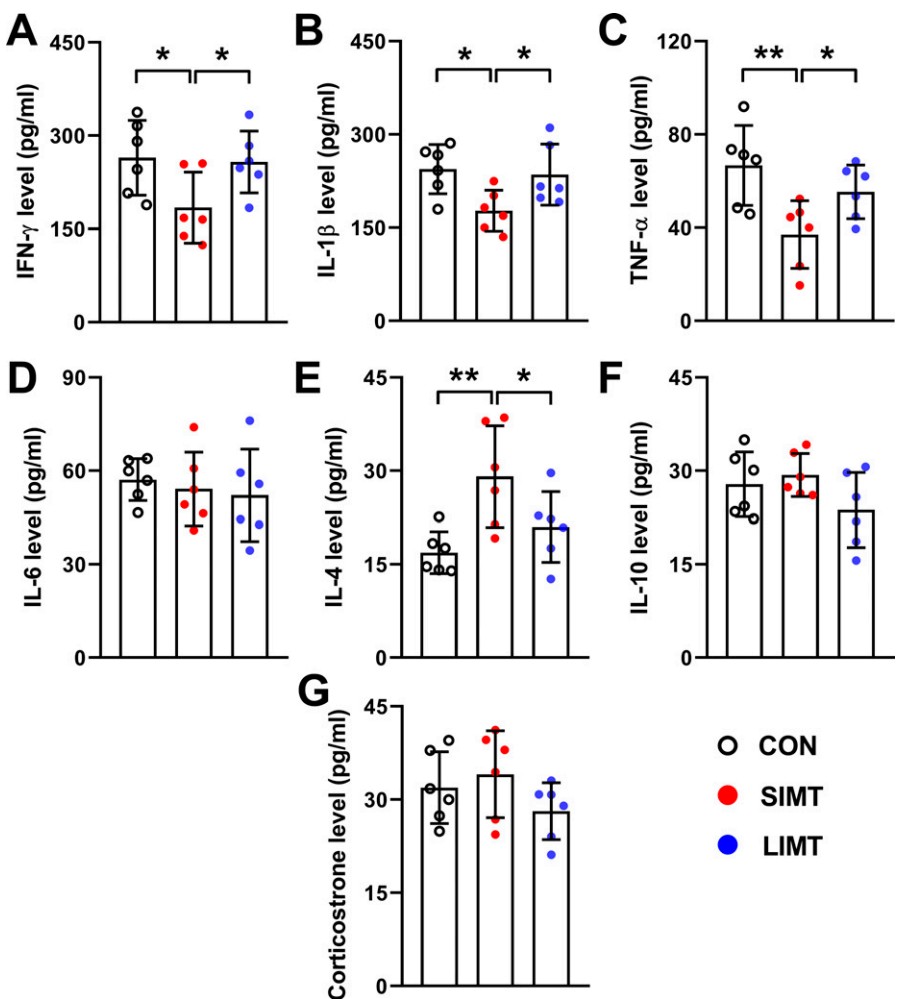

**Figure 9. Small intestinal microbiota transplantation under physiological conditions induced an anti-inflammatory cytokines profile in blood.**
**(A, B, C, D, E, F)** Bars represent mean value of the level of the detected cytokines in each of three groups.
**(G)** Bars represent mean value of the level of corticosterone in the blood. All data were analyzed using one-way ANOVA followed by LSD post hoc test. *$P < 0.05$; **$P < 0.01$. n = 6. Data were shown in figures as mean ± SD.

of Experiment Animal Center of Guangdong Pharmaceutical University and under conditions of constant temperature, humidity, and light (22 ± 2°C, 55% ± 5%, and 12-h light/dark cycle). The mice had access to the same formula of standard autoclaved feed and water ad libitum. All animals, either used for donors or for recipients, came from different dams so as to minimize influence exerted by the genetic factor. After shipped, all mice were housed individually in a sterile plastic cage (18 × 28 × 12 cm) with clean bedding in the same room. Health checks were carried out daily at light onset. Cage changes were done once every 3 d at the same time as regularly scheduled health checks. The animals were transferred into a sterile cage with clean padding by gently picking them up by the base of their tails with a gloved hand. All experiments were approved by the Guangdong Pharmaceutical University Animal Care and Use Committee.

This study consisted of two experiments. Experiment 1 aimed to test the effects of SIMT and LIMT on defecation habit, stool feature, fecal microbiota composition, and cytokines levels. Experiment 2 aimed to test that on depressive-like behaviors. 18 mice were used as recipients in Experiment 1 that were randomly assigned to the SIMT, LIMT, and CON groups (6 mice per group). 36 mice were used as recipients in Experiment 2 with the same grouping procedure (12 mice per group). Another 20 male C57BL/6 mice of the matched age

as recipients were used as donors. However, each of the animals was not selected randomly from their own littermates, they were rather selected according to these criteria: only male offsprings, high similarity in body weight (within the range: 23.55–23.65 g) of all selected mice, healthy whiskers, healthy hair (no injury, no falling off, no gray hair), healthy skin, healthy teeth, and healthy vision.

### Preparation of intestinal microbiota graft material

20 healthy male mice were used as the donor mice for both Experiment 1 and Experiment 2. On PND77, they were over-anesthetized with 10% chloral hydrate. In a sterile clean bench, the partial gastrointestinal tract was excised from the gastroduodenal junction to the anal sphincter and split into the small intestine and large intestine. Then all the following procedures were performed on the ice or at 4°C. The entire substance in the whole-length small intestine was flushed out using 2–5 ml of cold (4°C) PBS followed by very gentle squeezing with tweezers to avoid mucosal damage. In a sterile blender under 5% hydrogen, 10% carbon dioxide, and 85% nitrogen, the substance in the small intestine of all donor mice were mixed before homogenized. The particles were then removed by passing through stainless steel

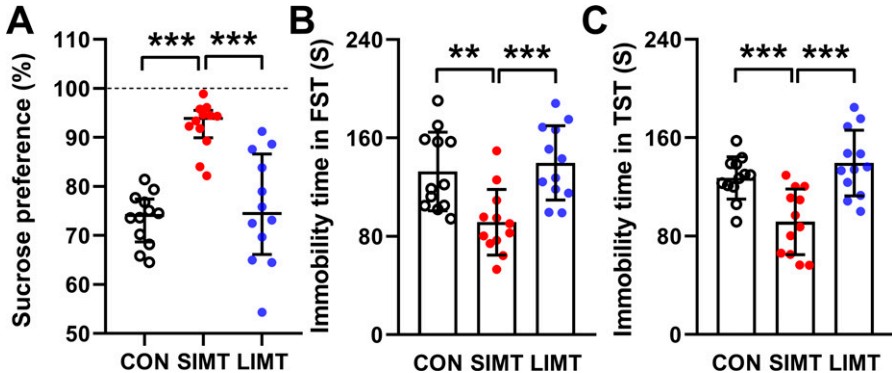

**Figure 10. Small intestinal microbiota transplantation under physiological conditions decreased depressive-like behaviors.**
**(A)** Data obtained in the sucrose preference test were shown as median ± interquartile in each of three groups. These data were analyzed by Kruskal–Wallis *H* test followed by Bonferroni post hoc test. ***$P < 0.001$. n = 12. **(B, C)** Bars represent mean value of the immobility time in forced swimming test (B) and tail suspension test (C). Data were analyzed using one-way ANOVA followed by LSD post hoc test. **$P < 0.01$; ***$P < 0.001$. n = 12. Data were shown in figures as mean ± SD.

laboratory sieves (0.25 mm-pore size; WS Tyler). The material was centrifuged (6,000*g*, 4°C, 15 min) (Staley et al, 2017). After discarding the supernatant, the remaining material was resuspended in sterile PBS. The resuspended SIMG sample was then divided into 140 aliquots, with every single aliquot being of a volume at 0.2 ml. One aliquot was used for quantifying the total bacteria number in a single aliquot microscopically using a Petroff-Hauser counting chamber (Wang et al., 1994, 2004) and another seven aliquots were pooled to be used for 16S rRNA gene sequencing. The rest 132 aliquots of samples were frozen in 10% (vol/vol) glycerol and stored frozen at –80°C until used (Hamilton et al, 2013). The same procedures were performed with the entire material in the whole-length larger intestine with another set of sterile tools for each animal. Before counting the bacteria in the small intestine microbiota graft sample, one aliquot (0.2 ml) sample was first diluted in 1:5 and then was divided into five equal aliquots (0.2 ml/aliquot). Each of the five equal aliquots was repeatedly used for counting using a Petroff-Hauser counting chamber. And the mean number was calculated from the five raw counting values. Before counting the bacteria in the large intestine microbiota graft sample, one aliquot (0.2 ml) sample was first diluted in 1:500 and then was divided into 100 equal aliquots (1 ml/aliquot). After trying to counting using one of the 100 aliquots, we found there were still too high concentration of bacteria in it to counting using a Petroff-Hauser counting chamber. So, we randomly selected 5 of the rest 99 aliquots (1 ml/aliquot) and further diluted them again in 1:100 before subjected to counting. Similarly, the mean number was calculated from the five raw counting values counting. And the mean number was calculated from the five raw counting values.

### SIMT and LIMT

SIMT and LIMT were performed from PND78. Each mouse in the SIMT group received seven oral gavages (once every other day) of SIMG that were thawed at room temperature before use. Each gavage used a single prepared aliquot that contained ~8.2 × $10^5$ bacteria cells. Likewise, each mouse in the LIMT group received seven oral gavages of thawed LIMG (once every another day). Each gavage used a single prepared aliquot that contained ~6.7 × $10^{10}$ bacteria cells. And each mouse in the CON group received seven oral gavages of 0.2-ml sterile PBS with the same procedure.

### Measurement of food intake, water intake, and body weight alteration

Each mouse was given a measured amount of food and water. Water and food intake were measured on PND75 (before microbiota grafts collection in donor mice and before microbiota transplantation/ sham procedure in all three groups of recipient mice) and on PND80, PND82, PND84, PND86, PND88, PND90, and PND92 (only in three groups of recipient mice during the two transplantation weeks). Water was measured by weighing the water and food intake was estimated according to the method previously described (Lin et al, 2000). Body weight of the 20 donor mice and the 18 recipient mice were measured at being ordered. Just before the donors were euthanized for graft collection, the body weight of each of them was measure again to evaluate the variation of their alterations in body weight within the several weeks from being ordered at P56. Body weight alterations of each of the 18 recipient mice were calculated from body weight data measured immediately before and 48 h after the finish of the whole microbiota transplantation/sham procedure.

### Analyses of number of stool pellets, stool wet weight, and stool water content

The number of stool pellets was measured on PND76 (before microbiota grafts collection in donor mice and before microbiota transplantation/sham procedure in all three groups of recipient mice). On PND92, it was also measured for the three groups of recipient mice. Specifically, stool pellets on each of the 2 d were collected for 6-h (from 11:00 AM to 17:00 PM). All pellets from one animal were put into a centrifuge tube and weighed to obtain a total wet weight. Then they were dried overnight at 65°C (Bengoa-Vergniory et al, 2020) and weighed again to test their dry weight. The stool water percentage was calculated from the difference between the wet and dry stool weights.

### Stool collection for microbiota analysis

Stool samples of the recipients were used for microbiota analysis because the microbiota component (e.g., the relative abundance of certain microbiota) of these samples represented the stable and final effects of microbiota transplantation on the gut microbiota composition of the recipients. From 8:00 AM to 11:00 AM on PND77 and PND92, mice were placed in the new sterile cages individually and the fresh stool pellets were

harvested. The stool pellets were homogenized into sterile PBS in sterile centrifuge tubes and then stored at –80°C until the DNA extraction.

## Blood sample collection and ELISA analysis

After the collection of stool samples, the mice were anesthetized deeply with 10% chloral hydrate before the blood was collected from the heart. The blood samples stored in tubes were left at room temperature for 30 min and then centrifuged (4,000g, 10 min, at 4°C) for serum separation. Then, the prepared serum samples were used strictly according to the manufacturer's protocols for ELISA assays to determine the levels of IFN-γ, TNF-α, IL-6, IL-4, IL-10, and corticosterone. Mouse IFN-γ (AN-18) ELISA set, Mouse IL-4 ELISA set, and Mouse IL-10 ELISA set were purchased from BD PharmingenTM (BD Biosciences). Mouse TNF-α ELISA kit, Mouse IL-6 ELISA kit, and corticosterone ELISA kit were purchased from EIAab Science Co, Ltd.

## Microbial profiling of grafts and fecal samples

Microbial DNA was extracted from the grafts and fecal samples using the Mag-Bind Soil DNA Kit (Omega Bio-tek) strictly according to the manufacturer's protocols. The bacteria 16S rRNA genes were amplified and then the resulting polymerase chain reaction products were extracted and purified. After this, purified amplicons were then pooled in equimolar concentrations and paired-end sequenced on an Illumina MiSeq platform (Illumina). The resulting data were analyzed on the Majorbio Cloud Platform (https://www.majorbio.com). The raw reads were deposited into the National Center for Biotechnology Information (NCBI) Sequence Read Archive (SRA) database (BioProject: PRJNA682764). See the following three paragraphs for details of DNA extraction and PCR amplification, Illumina MiSeq sequencing, and processing of sequencing data from the three recipient groups' samples before and after microbiota transplantation/sham procedure and the two grafts samples.

### DNA extraction and PCR amplification
Microbial DNA was extracted from the grafts and fecal samples using the Mag-Bind Soil DNA Kit (Omega Bio-tek) strictly according to the manufacturer's protocols. The concentration and purification of the final DNA extracted from each sample was quantified using NanoDrop 2000 UV-vis spectrophotometer (Thermo Fisher Scientific). DNA quality was assessed by 1% agarose gel electrophoresis. The V3-V4 variable regions of the 16S rRNA gene extracted from each sample were amplified by thermocycler PCR system (GeneAmp 9700; ABI) with primers 338F (5′-ACTCCTACGGGAGG-CAGCAG-3′) and 806R (5′-GGACTACHVGGGTWTCTAAT-3′). The PCR reactions were performed using the following thermocycling program: 3 min of denaturation at 95°C, 27 cycles (30 s at 95°C, 30 s for annealing at 55°C, and 45 s for elongation at 72°C), and a final extension at 72°C for 10 min. PCR reactions were conducted in triplicate 20 $\mu$l mixture including 4 $\mu$l of 5 × FastPfu buffer, 2 $\mu$l of 2.5 mM dNTPs, 0.8 $\mu$l of each primer (5 $\mu$M), 0.4 $\mu$l of FastPfu polymerase, and 10 ng of template DNA. The resulted PCR products were extracted from a 2% agarose gel and then further purified with the AxyPrep DNA Gel Extraction Kit (Axygen Biosciences) and finally quantified using

QuantiFluor-ST (Promega) according to the manufacturer's protocol.

### Illumina MiSeq sequencing
After the individual quantification step, amplicons were pooled in equal amounts, and pair-end 2 × 300-base pair (bp) sequencing was performed using an Illumina MiSeq platform (Illumina) according to the standard protocols by Majorbio Bio-Pharm Technology Co. Ltd. The raw reads were deposited into the NCBI SRA database (BioProject: PRJNA682764).

### Processing of sequencing data
Raw fastq files were demultiplexed, quality-filtered by Trimmomatic, and merged by FLASH with the following criteria: (i) the reads were truncated at any site receiving an average quality score <20 over a 50-bp sliding window; (ii) primers were exactly matched allowing two nucleotide mismatching, and reads containing ambiguous bases were removed; (iii) sequences whose overlap longer than 10 bp were merged according to their overlap sequence. Operational taxonomic units (OTUs) were clustered with 97% similarity cutoff using UPARSE (version 7.1 http://drive5.com/uparse/) and chimeric sequences were identified and removed using UCHIME. The taxonomy of each 16S rRNA gene sequence was analyzed by RDP Classifier algorithm (http://rdp.cme.msu.edu/) against the Silva (SSU138) 16S_bacteria rRNA database using confidence threshold of 70%.

## Behavioral tests

Mice in Experiment 2 were subjected to behavioral tests from PND92 to PND97. These behavioral tests consisted of SPT, FST, and TST, and all were performed according to the previous literature (Zhang et al, 2019).

### SPT
During PND92 to PND94, animals were allowed to habituate to drinking from one bottle of 1% sucrose solution and one bottle of water. On PND95, mice were given only water. On PND96, mice were subjected to a 24-h preference test in which water and 1% sucrose solution were delivered from identical bottles. The positions of the two bottles were switched every 4 h and measurement of water and sucrose intake were conducted every 24 h by weighing the bottles at the start and end of the testing period. The sucrose preference (%) was calculated as the volume of sucrose intake over the total volume of fluid intake.

### FST
On PND97, mice were allowed to adapt to the experimental environment for 1 h before testing. Then, the mouse was placed in a transparent cylinder (diameter: 20 cm; height: 40 cm), which was filled with water (23 ± 1°C) of a 15-cm depth. During the six-min task, the former 2 min were given to mice for adaptation and immobility time was formally counted by a video tracking system EthoVision (Noldus Information Technology B.V.) during the later 4 min. After each task, water was changed and the mouse was removed from the water, dried with tissues, and placed into a clean cage without the mouse ready to be tested to avoid communication (Slattery & Cryan, 2012).

### TST

Each mouse was subjected to TST task 4 h after finishing the FST task. Approximately 60 cm above the ground, each mouse was suspended upside down by taping its tail 1 cm away from the tip. Every 6-min trial was recorded by the camera right in front of the mouse. After a 2-min habituation period, the total immobility time was counted by a video tracking system EthoVision (Noldus Information Technology B.V.) during the later 4 min.

### Statistical analysis

Data reported in Figs 4–8 were analyzed on Majorbio Cloud Platform (https://www.majorbio.com). The rest data were analyzed statistically using SPSS Statistics software (Version 25.0; SPSS Inc.). Data were analyzed using $t$ test or one-way ANOVA followed by LSD post hoc test or Kruskal-Wallis $H$ test followed by the Bonferroni post hoc test. Significance was accepted at $P < 0.05$. Data were shown in figures as mean ± SD except for the data in Fig 6A which were shown as median ± interquartile.

## Data Availability

The datasets supporting the conclusions of this article are included within the article and its additional files. Raw 16S rRNA reads have been made available on the SRA under BioProject: PRJNA682764 (SAMN18961014 to SAMN18961031; SAMN17007146 to SAMN17007165).

## Supplementary Information

## Acknowledgements

We thank Ms Zhiqin Yang (Aviation health center, China southern airlines company limited, Guangzhou 51000, China) for her valuable discussion and help with this investigation. This work was supported by (the National Natural Science Foundation of China) under grant (No.31600836) and (the starting fund for high-level talent introduction into Guangdong Pharmaceutical University) under grant (No.51355093).

## Author Contributions

Y Xie: data curation, investigation, methodology, and writing—original draft, review, and editing.
L Song: data curation, formal analysis, investigation, and methodology.
J Yang: conceptualization, formal analysis, supervision, funding acquisition, writing—original draft, review, and editing.
T Tao: investigation and methodology.
J Yu: investigation and writing—review and editing.
J Shi: formal analysis and writing—review and editing.
X Jin: resources.

## Conflict of Interest Statement

The authors declare that they have no conflict of interest.

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
