## [Reviewer comments · Life Science Alliance]

Life Science Alliance

Small Intestinal Flora Graft Alters Fecal Flora, Stool, Cytokines and Mood Status in Healthy Mice

Yinyin Xie, Linyang Song, Junhua Yang, Taoqi Tao, Jing Yu, Jingrong Shi, and Xiaobao Jin
DOI: <https://doi.org/10.26508/lsa.202101039>

Corresponding author(s): Junhua Yang, Guangdong Pharmaceutical University

Review Timeline:

Submission Date:	2021-01-27
Editorial Decision:	2021-03-26
Revision Received:	2021-06-19
Editorial Decision:	2021-07-07
Revision Received:	2021-07-12
Accepted:	2021-07-14

Transaction Report:

March 26, 2021

Re: Life Science Alliance manuscript #LSA-2021-01039-T

Dr. Junhua Yang
Guangdong Pharmaceutical University
Department of Anatomy, School of Biosciences & Biopharmaceutics
Guangzhou, Guangdong 510006
China

Dear Dr. Yang,

Thank you for submitting your manuscript entitled "Transplanting small intestinal flora benefits fecal flora, defecation, cytokines and mood status in healthy mice" to Life Science Alliance. The manuscript was assessed by expert reviewers, whose comments are appended to this letter. We would like to invite you to submit a revised manuscript to LSA that addresses all of the reviewers' points.

We apologize for this unusual and extended delay in getting back to you. As you will note from the reviewers' comments below, while overall the reviewers do find the study interesting, they have also raised a number of significant questions and concerns, all of which must be addressed in the revised manuscript. Strong reviewer support from both reviewers will be required at re-review, and in particular, more detailed and precise information about donor mice must be included in the revised manuscript.

Thank you for this interesting contribution to Life Science Alliance. We are looking forward to receiving your revised manuscript.

Sincerely,

Shachi Bhatt, Ph.D.
Executive Editor
Life Science Alliance
<https://www.lsjournal.org/>
Tweet @SciBhatt @LSAJournal

- A letter addressing the reviewers' comments point by point.
- An editable version of the final text (.DOC or .DOCX) is needed for copyediting (no PDFs).
- High-resolution figure, supplementary figure and video files uploaded as individual files: See our detailed guidelines for preparing your production-ready images, <https://www.life-science-alliance.org/authors>
- Summary blurb (enter in submission system): A short text summarizing in a single sentence the study (max. 200 characters including spaces). This text is used in conjunction with the titles of papers, hence should be informative and complementary to the title and running title. It should describe the context and significance of the findings for a general readership; it should be written in the present tense and refer to the work in the third person. Author names should not be mentioned.

B. MANUSCRIPT ORGANIZATION AND FORMATTING:

Reviewer #1 (Comments to the Authors (Required)):

The study is an interesting observational study which demonstrates that transplant with either small intestinal or large intestinal microbiota transplant in a similar post-transplant graft profile when transferred into recipient mice. However, divergent effects on the host in the context of physiology,

circulating cytokine profile and behaviour were observed depending on whether mice received small or large intestinal transplants. These data are observational only.

There are some obvious omissions of groups which should be added to the Community heatmap analysis on OTU level and to the PCoA on OTU level, namely the grafts for each recipient animal and their own microbiota before and after transplant, at least at faecal level.

However, my biggest concern is the lack of information on the donor mice. These mice appear to have been a different cohort from the recipient mice. Ideally, the donors, to ensure some degree of microbiome homogeneity, should have been either littermates or cage-mates of the recipient mice. Moreover, we do not know what the behavioural profile of the donor mice was. It is also unclear how the stool microbiome profile of the recipient mice changed before and after transplant. There is also no control small intestinal microbiome profile.

Methodologically, many studies prepare rodents, and human subjects, with either a 'wash out' or antibiotic treatment to prepare the host to receive the transplant. This does not appear to be the case in this study, therefore the final microbiota profile likely reflects some mix of the host and transplant from an unrelated donor. Donor health status and any contaminant in the small intestine in particular could account for the data observed. The facility was only SPF, but ideally, as noted, the donor and recipient mice should really have been litter- or cage-mates. Much more detail on housing conditions should also be provided both before and after transplant - cage effects could be a significant issue in this study depending on housing conditions.

Reviewer #2 (Comments to the Authors (Required)):

Review for article entitled "Transplanting small intestinal flora benefits fecal flora, defecation, cytokines and mood status in healthy mice"

This study aims to assess the influence exerted by the small intestinal microbiota compared to the large intestinal microbiota on their host. The authors compare the physiological, immune, endocrine and behavioural indicators following the transplantation of small intestinal flora, large intestinal flora or PBS from healthy adult mice to age-matched healthy mouse recipient. In this study, the authors find significant alterations in defecation features, fecal microbial composition, cytokines profile and depressive-like behaviour following small intestinal microbiota transplantation. In contrast, large intestinal microbiota transplantation resulted in only slight alterations in the fecal microbiota composition and did not influence other factors. Thus, the author concludes that the microbiota from the large intestinal microbiota and small intestinal microbiota exert different functions. In my opinion, this study will increase the cumulative knowledge of the intestinal microbiota and is worthy of publication in Life Science Alliance if the following issues are addressed:

Major concerns:

Line 124-126: Why was there no comparison done between the two experimental groups (SIMT and LIMT) as well as a comparison including the control feces and SIMT group to cover all bases?

Line 124-126: Please provide rationale as to why microbial composition was not evaluated for LIMG.

Line 162-163: Given that feces is primarily components of the large intestine and the microbiota between the donor group and the recipient group were not drastically different, would it not be expected to see only a slight change in the intestinal microbiota given LIMG? Whereas in the SIMG group a drastically different microbiome is administered explaining the significant changes.

Moreover, if microbiota in the large intestine was drastically different (similar to most cases where FMT is administered) results could be more similar to the results seen by transplantation of the

small intestine microbiota.

Line 217-220: Although OTU level analysis provides a reasonable level of more detail at the genus level, use of amplicon sequence variants (ASVs) would provide a far more precise analysis at the species level. Given the wide availability of denoising software capable of the latter, one would expect an ASV approach to be implemented rather than an OTU approach. Please provide justification for using this out-dated strategy.

Line 355-356: Why was quantity of stool pellets not measured before mice were given SIMG or LIMG.

Line 319: Using only one aliquot to determine the total bacteria number does not leave room for error. Should this not have been done in biological or at least technical replicates?

Line 197-199: Could these changes observed with SIMT be the result of a lack of diversity between the small and large intestine? Perhaps in SIMT, engraftment occurred at the small intestine because the composition of SIMG was similar to the small intestinal environment, whereas LIMT engraftment occurred only in the large intestine

Minor Concerns:

Figure 1: It is clear through plots on grafts A-D, that eating and drinking habits varied between the individual mice in each group. Why were these factors were not considered before transplantation given that the variations between individuals could expectedly impact the results.

Line 78-79: Please provide a reference for the sentence "compared with microbiota in the small intestine, microbiota in the large intestine is more similar to fecal microbiota."

Line 304-325: Please clarify whether the donor microbiota was pooled before homogenization or if each donor was homogenized individually.

Line 316: It is unclear if centrifugation was done performed on the particles removed in 312 or the filtered substance.

Line 338-339: Why was the body weight recorded for the LIMT and SIMT group but not the control group? Your results state otherwise, please indicate that these measure were also done for your control group.

These results show that only change was seen in the large intestine given FMT, did the small intestine change? If the small intestine closely matched the composition of the LIMT than the idea that the microbiota belonging to the two environments exert different functions would not be correct.

Responses to Reviewer #1

The study is an interesting observational study which demonstrates that transplant with either small intestinal or large intestinal microbiota transplant in a similar post-transplant graft profile when transferred into recipient mice. However, divergent effects on the host in the context of physiology, circulating cytokine profile and behavior were observed depending on whether mice received small or large intestinal transplants. These data are observational only.

Question-1

There are some obvious omissions of groups which should be added to the Community heatmap analysis on OTU level and to the PCoA on OTU level, namely the grafts for each recipient animal and their own microbiota before and after transplant, at least at fecal level.

Response-1

Thank the reviewer for this professional comment. In fact, we had analyzed fecal microbial composition before transplantation at the OTU level in three groups of recipient mice, with no significant between-groups differences found. Moreover, there was also no significant difference in fecal microbial composition between the fecal samples from the control group before and after the post-sham transplantation treatment. Hence, we thought that it was unnecessary to report the data from fecal samples collected before microbiota transplantation.

In addition, the health status indexes including food intake, water consumption, body weight, stool pellets number, stool wet weight and stool water percentage of both the donor mice and the recipient mice before microbiota transplantation had also been recorded and analyzed with no significant differences found between three recipient groups or between the donor and the recipient mice. Therefore, we also thought that it was unnecessary to report these data.

However, after studying this piece of professional comment, we have realized that we should have provided these data in our initial submission. Now, we have added all these data to the revised manuscript accordingly (see section 2.1-2.2, Fig.1, Fig2 for data of health status indexes before microbiota transplantation. Line100-123) (see section 2.5 and Fig.4 for analysis of microbial community composition between three groups of recipient mice before microbiota transplantation. Line141-151) (see section 2.6 and Fig.5 for analysis of microbial community composition before and after microbiota transplantation. Line152-164). The corresponding method information has also added to the revised version (Line421-423, Line427-433 and Line435-437 for data of health status indexes before microbiota transplantation) (Line447 for analysis of microbial community composition before microbiota transplantation).

Because the substance in the small intestine of all donor mice were mixed and homogenized before being divided into 140 aliquots with the same volume (0.2 mL)

and the same procedure was performed with the substance in the large intestine of all donor mice, the graft samples received by each of the recipient animals within a group were of the same microbial composition. Therefore, the grafts for each recipient animal were not analyzed for their microbial composition. Instead, a total volume of 1.4 mL graft sample (equal to seven aliquots, Line394-395) was used as a representative sample in the analysis for the microbial composition contained in small or large intestine microbiota graft.

Question-2

However, my biggest concern is the lack of information on the donor mice. These mice appear to have been a different cohort from the recipient mice. Ideally, the donors, to ensure some degree of microbiome homogeneity, should have been either littermates or cage-mates of the recipient mice.

Response-2

As the reviewer's professional comment pointed out, using the recipients' littermates as donors could ensure of microbiome homogeneity between the recipients and the matched donors within each litter, which help to minimize the influence by the heterogeneity of the microbiome between the recipients and the donors within each litter. Therefore, we had considered to use such a design before starting this study.

However, other questions happen to such a design. Because each of the donors come from different litters even in such a design, the donors provided grafts certainly with different microbial community composition and thus the recipients from different litters were given different grafts. In other words, recipients actually received different treatment within the same treatment group, such as the small intestine graft transplantation group. Such a design might also, from this perspective, to impair the homogeneity of the microbiome in the grafts. This would undoubtedly lead to an additional variability among the recipient mice within the same treatment group.

Additionally, not only the donor mice have been a different cohort from the recipient mice, but each of the donor mice and recipient mice come from different dams/litters. One may ask that since the donors were set as described in our manuscript (an independent cohort, come from different dams/litters and mixed all their substances in their small intestine or that in large intestine to make the grafts homogenetic), why the authors did not set the recipients in a randomized complete block design with litter as the block factor and microbiota transplantation as the treatment factor.

A randomized complete block design is better to minimize the effects by some non-treatment factors, especially genetic background. But the statistician we have consulted before starting this study (Jingrong Shi, who also participated in the data analyses and other important work in revising this manuscript and thereby has been added to the author list, at Department of Data Mining and Analysis, Guangzhou

Tianpeng Technology Co.,Ltd.) told us that a randomized complete block design was not proper for our study for at least the four reasons here:

1) Sex factor. Female C57Bl/6 mice are more susceptible to stress-induced neurobehavioral alterations, such as dysregulation of the HPA axis, whereas male C57Bl/6 mice were reported to be of more resilience [PMID: 30450373] [PMID: 26674863][PMID: 32154650]. Given that the current study aimed at comparing the effects of SIMT and LIMT in healthy host or under physical condition, only male C57Bl/6 mice thus were used in our study.

Typically, no more than six, and often three to five, single sex pups were delivered to one pregnant mouse. So, it would be difficult to ensure replicates of animals for each of the three treatments (sham transplantation procedure, SIMT or LIMT) within one litter. If such replicates were set as two animals, eight single sex pups within one litter would be required, that is, two were used for CON group, two for SIMT group, two for LIMT group and the rest two were used as litter-matched donor mice.

2) Development status. All selected animals, including recipient and donor mice, were of high similarity in body weight (within the range: 23.55-23.65g) so as to the largest extent to avoid the influences by the differences of individuals in their development status. Littermates often had obvious differences in their development status due to prenatal, maternal caring and lactation differences.

3) Injuries by territorial behavior of male mice or by other factors. All selected animals, including recipient and donor mice, were of healthy whiskers, healthy hair (no injury, no falling off, no gray hair) and healthy skin. In addition, territorial behavior of the male littermates would inevitably result in difference in mood status, reactivity to stress, etc.

4) Other healthy status evaluated by non-invasive methods. All selected animals, including recipient and donor mice, were of healthy teeth and healthy vision. In fact, malocclusion followed by acquired progressive dystrophy often happens to C57Bl/6 mice [PMID: 26876137]. Moreover, about 12% C57Bl/6 mice had eye defects [PMID: 10393064].

In a word, these reasons would make the sample size and the homogeneity in non-treatment aspects of recipient animals insufficient to the requirements in a randomized complete block design. Now, we have added these details to section 4.1 to clarify the animals' information (Line356-363).

The problems of territorial behavior of male mice also exist in the case that cage-mates were used for donor and/or recipient mice, just as in case of littermates discussed as above.

Question-3

Moreover, we do not know what the behavioral profile of the donor mice was.

Response-3

Thanks for the reviewer' comment. To know what the behavioral profile of the donor mice was would indeed help analyze the possible difference before transplantation between donor and recipient mice. However, performing FST and TST tasks could led to a release of corticosterone in mice [PMID: 20869223]. FST and TST are actually the most widely used methods for modeling depression like behavior in rodents [PMID: 20472130]. Therefore, testing the mood related behaviors would in itself serve as a stress and exert influences on the the mood related behaviors such as anxiety and depression like behaviors. Such alterations in stress hormone and mood status may result in some change in microbial community composition and even may be adoptively transferred through microbial transplantation to the recipient mice [PMID: 33311466], which would interfere the aimed observation of our study. Hence, the behavioral profile of the donor mice was not measured. Nevertheless, the similarity of the physical status between donor and recipient mice was well ensured not only by the identical nature in sex, age, strain and producer, but also by the high similarity in body weight and other health status as stated in section4.1 in the maintext (Line351-353, Line356-358, Line372-376).

Question-4

It is also unclear how the stool microbiome profile of the recipient mice changed before and after transplant.

Response-4

Thank the reviewer for this professional comment. As stated in Response-1, we had analyzed fecal microbial composition before transplantation at the OTU level in three groups of recipient mice.

Now, these data have been reported in the current version of our manuscript. As expected, SIMT group showed a significant different profile after transplantation compared with before it. There was also no significant difference in fecal microbial composition between the fecal samples from the LIMT group before and after transplantation. (see section 2.6 and Fig.5 for analysis of microbial community composition before and after microbiota transplantation. Line152-164)

Question-5

There is also no control small intestinal microbiome profile.

Response-5

We are very sorry for not completely understanding the exact meaning of this piece of comment. By saying this comment, we think, the reviewer may mean that microbiome profile in the small intestine was not analyzed between three recipient groups after transplantation, or that microbiome profile in the small intestine of each of the three recipient groups was analyzed between before and after transplantation. In short, we did not analyze the possible alterations in the small intestinal microbiome profile either between the recipient groups or within each certain recipient group.

Since the fecal microbiome profile was altered by transplantation, it is very possible that the microbiome profile in small intestine and large intestine of recipient mice also altered after transplantation. And therefore, testing the possible small intestinal microbiome profile is likely to provide more detailed information about the effects of transplantation on gut microbial community composition.

However, microbial community composition in small intestine samples possesses much larger variation than that in fecal or large intestine samples [PMID: 24116019] [PMID: 24592323], may due to the surrounding bacteria intake along food/water intake in the SPF housing condition or due to the oral microbiota downstream along the gastrointestinal tract [PMID: 24116019]. The much larger variation of small intestinal microbiome profile lessens the significance of this index in stably reflecting the influence of transplantation on gut microbiota.

In fact, to our best knowledge, rare study reported the small intestinal microbiome profile after fecal/gut microbiota transplantation in human or rodents, also suggesting analyzing small intestinal microbiome profile might be of smaller value in investigating the influence of transplantation on gut microbiota.

On the contrary, a great number of studies carried out microbial analyses in fecal samples after fecal/gut microbiota transplantation in human or rodents. So, the data of fecal samples are easier than small intestinal samples to be compared with the results reported by previous studies concerning fecal/gut microbiota transplantation in healthy/control/wild-type mice.

Moreover, it has been verified in pig [PMID: 30563199] that fecal sample, rather than small intestinal sample, reflects better the stable influence of microbiota transplantation.

All these reasons made us decide to analyze fecal sample and not to analyze small intestine of the recipient mice either before or after transplantation to evaluate the differences between SIMT and LIMT.

Question-6

Methodologically, many studies prepare rodents, and human subjects, with either a 'wash out' or antibiotic treatment to prepare the host to receive the transplant. This does not appear to be the case in this study, therefore the final microbiota profile likely reflects some mix of the host and transplant from an unrelated donor.

Response-6

Admittedly, many studies prepare rodents, and human subjects, with either a 'wash out' or antibiotic treatment to prepare the host to receive the transplant. As far as we know, these studies were designed to investigate the role of gut microbiota in the pathology of some disease or the therapeutic role of gut microbiota for some disease [PMID: 30319571][PMID: 29083037][PMID: 30920075][PMID: 25271725]. In these studies, the gut microbiota were 'washed out' or treated with antibiotics before transplantation in order to replace completely the inherent gut microbiota with the graft-containing microbiota. This replacement undoubtedly facilitates the investigation of the role of gut microbiota in pathogenesis and/or treatment of some disease.

Our present study aimed to address a different issue, that is, to explore whether healthy hosts that harbor completely normal gut microbiota are differently influenced by small intestine microbiota and large intestine microbiota transplantation that are also from healthy mice. The hypothesized difference between small and large intestine microbiota transplantation in physical condition, if be verified, would theoretically serve as a basic mechanism underlying any potential similar different roles of small and large intestine microbiota transplantation in various pathological conditions. This is why we designed the present study using healthy donor and recipient mice without a 'wash out' or antibiotic treatment to prepare the host to receive the transplantation.

A 'wash out' or antibiotic treatment not only are usually adopted to study disease-related role of gut microbiota, they in themselves also induce a non-healthy/imbalance/dysregulation condition both in gut microbiota profile and in the whole organism. As reviewed and concluded by, antibiotics overuse can drive to a series of harmful clinical consequences, including clostridium difficile infection, IBS, IBD, metabolic disorders or liver disease [PMID: 27531828] When the administration of antibiotics is stopped, the functions of intestinal microenvironment, endocrine system and immune system are all in the process of drastic changes. It is possible for the rebound proliferation in a non-physical abundance proportion to occur rapidly by the residual microorganisms in the digestive tube (although it may be very small)[PMID: 32817384]. It is also possible for the microorganisms in the surrounding environment to re-colonization into the digestive tract immediately after ceasing to use antibiotics [PMID: 12076027].

This unnatural and unstable state induced by a 'wash out' or antibiotic treatment, just like with physical gut microbiota of the host in the absence of 'wash out' or antibiotic treatment, will also interact with the graft, resulting in some mixed effects. The transplantation of small and large intestinal microbiota in case of a 'wash out' or antibiotic treatment was carried out first, it is not so much to observe the effect of the graft in the physical state, but to observe some unknown therapeutic effect in the state of intestinal bacterial imbalance or immune/endocrine dysfunction.

In this study, the normal gut bacteria of healthy recipients, as an inherent part of the healthy body as intestinal immune cells, epithelial cells and other physical components, may be affected by transplantation. This is supported by the data we collected about the changes of fecal microbial composition and immune factors in recipient mice after transplantation.

Indeed, each individual had its own unique genetic background and housing-cage, which may enlarge the heterogeneity of their gut microbiota composition, like the more or less heterogeneity in their immune system activity and endocrine system function. The variations resulting from these heterogeneities was put in the random error within treatment group and was distinguished from the effects of treatment between groups in the statistical analyses of our data. Hence, the results and conclusion of our manuscript are convincing even though a 'wash out' or antibiotic treatment was not carried out before transplantation.

Question-7

Donor health status and any contaminant in the small intestine in particular could account for the data observed. The facility was only SPF, but ideally, as noted, the donor and recipient mice should really have been litter- or cage-mates. Much more detail on housing conditions should also be provided both before and after transplant - cage effects could be a significant issue in this study depending on housing conditions.

Response-7

Thank the reviewer for his/her comment. As we understand, this piece of comment pointed out three issues: **1)** if the donors were of a different health status from the recipients, the donors' health status may, at least partially, account for the different effects between SIMT and LIMT; **2)** there may be some contaminant in the small intestine of the donors that could account for the data observed; **3)** in the non-germ free rearing/housing/experimental (SPF) environment, the litter- or cage-mates matched donor and recipient design was not used, so the authors should provide more information about housing condition and justify the rationality of the design and results of this study.

About the first issue: **Donor health status could account for the data observed.**

In fact, as stated in the **Response-2**, the initial health status of both donors and recipients and the highly homogeneity in health status between the donors and recipients before being ordered were guaranteed by a series of criteria, including high similarity in body weight (within the range: 23.55-23.65g), healthy whiskers, healthy hair, healthy skin, healthy teeth and healthy vision. In addition, as we described in the **Response-1**, the health status indexes including body weight, food intake, water consumption, stool pellets number, stool wet weight and stool water percentage of both the donor mice and the recipient mice before microbiota transplantation had also

been recorded and analyzed, with no significant differences found between three recipient groups or between the recipient and the donor mice.

Given that all mice used in our study were administrated without any treatment which may impair/interfere their health status, for instance, drug administration, surgical procedure, genetic knockout, stress-inducing behavioral tasks, etc., these results of general development, food/water intake, stool features reasonably showed that the donors had no different health status from the recipients. This thus suggests that the health status of the donors should not be the source of the post-transplant effect in the recipient mice.

The mood status and immune system status of donor mice also matter because these aspects were evaluated in the recipient mice after transplantation. However, we did not observe these aspects of the donor mice. As stated in the **Response-3**, testing the mood related behaviors would in itself serve as a stress and exert influences on the mood related behaviors such as anxiety and depression like behaviors. Such alterations in stress hormone and mood status may result in some change in microbial community composition and even may be adoptively transferred through microbial transplantation to the recipient mice [PMID: 33311466], which would interfere the aimed observation of our study.

The immune system status of donor mice was not tested because grabbing and blood collection operation also would lead to additional injuries, including psychological stress, hurt, inflammation even slight and transient. Moreover, the recipient mice ought to be subjected before transplantation to the same tests (cytokines levels tests in our study) if these tests were conducted for the donor mice, so as to analyze whether there was any possible difference between the donor and recipient mice in their immune system status and thus to allow us to know whether the donor immune system status matters for the transplantation-induced effects in recipient mice. It is not proper to test the immunological indexes of the recipient mice before transplantation for the same reason as the first sentence says of this paragraph.

Nevertheless, the similarity of the physical status between donor and recipient mice was well ensured not only by the identical nature in sex, age, strain and producer, but also by the high similarity in body weight and other health status as stated in section4.1 in the maintext (Line351-353, Line356-358, Line372-376).

About the second issue: **Any contaminant in the small intestine in particular could account for the data observed.**

All our experiments were carried out strictly in a standard experimental environment and with the operating rules abided strictly. All the mice were supported only by a standard commercial sterilized food and sterilized distilled water. Therefore, in theory, the mice had no chance to be exposed to any contaminant.

Besides, even if there is some unknown contaminant in the housing environment or diet and it enters the small intestine of the donor mice, it is also difficult for this contaminant to be responsible for the transplant-inducing effects in the recipient mice that we have observed. Because all mice, including donor and recipient mice, received the identical housing condition and diet. Another reason lies in the methodology, the preparation of gut microbiota grafts from donors involves mixing the substance from donors' intestine tract, diluting homogenization, resuspending/washing bacterial components with sterile PBS. Therefore, even if there is a certain contaminant in the small intestine of the donor mice, its residue in the eventually prepared graft samples should be very low, at least much lower than its level in the small intestine of the recipient mice taken through the same housing conditions and diet.

About the third issue: **About the housing conditions and cage effects.**

As we stated in detail in the **Response-2**, a design using litter- or cage-mates matched donor and recipient was not proper for our present study. Instead, all animals, either used for donors or for recipients, came from different dams so as to minimize influence exerted by genetic factor. What's more, all mice were housed individually in a sterile plastic cage (18 × 28 × 12 cm) with clean bedding in the same room. There is only one between-subjects independent variable in this design: treatment. This design ensures that the observed effects between recipient groups derive from different kinds of transplantation treatment, rather than from litter or cage factor.

We have now added the information about housing conditions to the section 4.1 of the main text of this manuscript according to this professional comment (Line 356-363).

Responses to Reviewer #2

In my opinion, this study will increase the cumulative knowledge of the intestinal microbiota and is worthy of publication in Life Science Alliance if the following issues are addressed:

Question-1

Line 124-126: Why was there no comparison done between the two experimental groups (SIMT and LIMT) as well as a comparison including the control feces and SIMT group to cover all bases?

Response-1

Sentences in the Line 124-126 in the maintext of the initially submitted version are '*Cluster 1 included the large intestinal microbiota graft sample (LIMG) and all fecal samples from the CON and LIMT groups. Cluster 2 included the SIMG and all fecal samples from the SIMT group*'. The two sentences state the results of clustering analysis of the microbial data from both *small and large intestinal microbiota graft samples (SIMG and LIMG)* and fecal samples from the CON, SIMT and LIMT groups after transplantation.

As stated in a literature, clustering analysis is the process of grouping the data into classes or clusters, so that objects within a cluster have high microbiota composition similarity in comparison to one another but these objects are very dissimilar to the objects that are in other clusters [A Study of Hierarchical Clustering Algorithm. International Journal of Information and Computation Technology. Volume 3, Number 10 (2013), pp. 1115-1122].

This clustering analysis results shown in Fig.2 in the initially submitted version (that is, Fig.4 in the revised version) tell that there are two first-level clusters that are described by the two sentences (Line 124-126 in the maintext of the initially submitted version) about which samples each of the two first-level clusters contains, so as to show which samples have greater similarity in this experiment.

As shown in the figure presented as follow, *Cluster 1 included the large intestinal microbiota graft sample (LIMG) and all fecal samples from the CON and LIMT groups*, suggests that large intestinal microbiota transplantation induced no significant shift of fecal microbiota profile or no distinguish fecal microbiota profile in LIMT group from that of the CON group. *Cluster 2 included the SIMG and all fecal samples from the SIMT group*, suggests that small intestinal microbiota transplantation induced a significant shift of fecal microbiota profile towards the microbial composition of the small intestinal microbiota graft (SIMG) sample or a distinguish fecal microbiota profile in SIMT group from that of the CON group.

In sum, the two sentences (Line 124-126 in the maintext of the initially submitted version) are not a description of some kind of inter-group comparison. We are very

sorry that our description might cause the reviewer's misunderstanding of these two sentences.

Question-2

Line 124-126: Please provide rational as to why microbial composition was not evaluated for LIMG.

Response-2

As stated in **Response-1**, the two sentences in Line 124-126 in the maintext of the initially submitted version were only the content describing the clustering analysis results shown in Fig.2 in the initially submitted version (that is, Fig.4 in the revised version), telling that there are two first-level clusters and each of them consists of which samples. Microbial composition was indeed evaluated for both small and large intestinal microbiota graft samples (SIMG and LIMG) as well as fecal samples from the CON, SIMT and LIMT groups after transplantation. It is the microbial data that were subjected to the clustering analysis as mentioned above.

Question-3

Line 162-163: Given that feces is primarily components of the large intestine and the microbiota between the donor group and the recipient group were not drastically different, would it not be expected to see only a slight change in the intestinal microbiota given LIMG? Whereas in the SIMG group a drastically different microbiome is administered explaining the significant changes. Moreover, if microbiota in the large intestine was drastically different (similar to most cases where FMT is administered), results could be more similar to the results seen by transplantation of the small intestine microbiota.

Response-3

We thank the reviewer for his/her professional comment that feces is primarily components of the large intestine and the microbiota between the donor group and the recipient group were not drastically different.

It has been reported [PMID: 24116019] that ‘the microbial profile in large intestine was more similar to feces than those in the small intestine, with the similarity of 0.75 and 0.38 on average, respectively’. A similar report has also been published [PMID: 25688558]. This finding supports the reviewer’s comment that feces is primarily components of the large intestine.

In addition, the similarity of the physical status between donor and recipient mice was well ensured not only by the identical nature in sex, age, strain and producer, but also by the high similarity in body weight (within the range: 23.55-23.65g at PND56), healthy whiskers/hair (no injury, no falling off, no gray hair) and/skin, healthy teeth and healthy vision (Line351-353, Line356-358, Line372-376). The health status of whiskers/hair/skin was checked because injuries in them often happen due to territorial behavior of male mice. The health status of vision and teeth were checked because eye defects and malocclusion followed by acquired progressive dystrophy often happens to C57Bl/6 mice [PMID: 26876137] [PMID: 10393064]. According to this well ensured similarity of the physical status between donor and recipient mice, the microbiota between the donor group and the recipient group should not be drastically different as the reviewer pointed out in this comment.

However, the microbial composition in the healthy large intestine is more similar to the microbial composition of the large intestinal microbiota graft sample (LIMG) than to that of the small intestinal microbiota graft sample (SIMG). Therefore, we deduce, it would only be expected to see mere a slight change in the intestinal microbiota given LIMG if the both bacteria in SIMG and that in the LIMG were administrated into the large intestine rather than into the stomach of the recipient mice. On the contrary, if the bacteria in SIMG and LIMG were administrated into the upper section of the gastrointestinal tract such as the stomach as in our present study, it would be more likely to see mere a slight change in the intestinal microbiota given SIMG and to see a significant change in the intestinal microbiota given LIMG because the microbial composition in the healthy small intestine is more similar to the microbial composition of the SIMG than to that of the LIMG.

But, this deduction, especially the second point, is not consistent either with the reviewer's comment or with our actual findings that SIMG, not LIMG, performed by intragastric administration induced a significant change in intestinal microbiota, defecation, circulating cytokines and mood status in healthy recipient mice. This inconsistency may derive from the complex and unrevealed competition and interaction between the microorganisms in the graft and the inherent flora in the digestive tract of the host, as well as the interaction between the microorganisms in the graft and the host immune system. In other words, it may be not proper to deduce the effects on intestinal microbiota composition by transplantation simply according to similarity of the microbiota composition in the graft and that in the host digestive tract.

We did not analyze the possible alterations in the small intestinal microbiome profile of the recipient groups, although testing it may provide more detailed information about the effects of transplantation on gut microbial community composition. The microbial community composition in small intestine possesses much larger variation than that in fecal or large intestine [PMID:24116019] [PMID:24592323], may due to the surrounding bacteria intake along food/water intake in the SPF housing condition or due to the oral microbiota downstream along the gastrointestinal tract [PMID:24116019]. This fact lessens the significance of this index in stably reflecting the influence of transplantation on gut microbiota. To our best knowledge, rare study reported the small intestinal microbiome profile after fecal/gut microbiota transplantation in human or rodents, also suggesting analyzing small intestinal microbiome profile might be of smaller value in investigating the influence of transplantation on gut microbiota. On the contrary, a great number of studies carried out microbial analyses in fecal samples after fecal/gut microbiota transplantation in human or rodents. So, the data of fecal samples are easier than small intestinal samples to be compared with the results reported by previous studies concerning fecal/gut microbiota transplantation in healthy/control/wild-type mice. Moreover, it has been verified in pig [PMID: 30563199] that fecal microbial sample reflects better the stable influence of microbiota transplantation than small intestinal microbial sample.

Question-4

Line 217-220: Although OTU level analysis provides a reasonable level of more detail at the genus level, use of amplicon sequence variants (ASVs) would provide a far more precise analysis at the species level. Given the wide availability of denoising software capable of the latter, one would expect an ASV approach to be implemented rather than an OTU approach. Please provide justification for using this out-dated strategy.

Response-4

Thank the reviewer for his/her professional comment. As reported by a study published in Oct. 2020, metabarcoding analysis has recently shifted from clustering

reads using Operational Taxonomical Units (OTUs) to Amplicon Sequence Variants (ASVs) [PMID: 33092529]. Differences between these methods can seriously affect the biological interpretation of metabarcoding data, especially in ecosystems with high microbial diversity, as the methods are benchmarked based on low diversity datasets [PMID: 33092529] .

In this reported work [PMID: 33092529], the authors have thoroughly examined the differences in community diversity, structure, and complexity between the OTU and ASV methods and argued that '*only when sequencing depth was high enough (> 50,000 sequences), the ASV method outperformed the OTU method in the samples with higher community richness*'. They also concluded that '*The ASV method used outperformed the OTU method, but OTU method is adequate to capture the community complexity*'.

Approximately half of the data given by 16S rRNA gene sequencing in our study have less than 50,000 sequences. Therefore, OTU method was selected according to this recent literature. We also notice that OTU method has been often used in studies published in major journals concerning microbiota analyses, such as *Geome Med* [PMID: 33658065]; *ISME J* [PMID: 32366970]; *J Pineal Res* [PMID: 32969515]; *Microbiome* [PMID: 33198805]; *Water Res* [PMID: 33126006]. Therefore, it should be proper that OUT method is used for the analyses in our current study.

Question-5

Line 355-356: Why was quantity of stool pellets not measured before mice were given SIMG or LIMG.

Response-5

Thank the reviewer for this professional comment. In fact, the health status indexes including body weight, food intake, water consumption, stool pellets number, stool wet weight and stool water percentage of both the donor mice and the recipient mice before microbiota transplantation had also been recorded and analyzed with no significant differences found between three recipient groups or between the recipient and the donor mice. Therefore, we thought that it was unnecessary to report these data.

In addition, we had analyzed fecal microbial composition before transplantation at the OTU level in three groups of recipient mice, with no significant between-groups differences found. Moreover, there was also no significant difference in fecal microbial composition between the fecal samples from the control group before and after the post-sham transplantation treatment. Hence, we also thought that it was unnecessary to report the data from fecal samples collected before microbiota transplantation.

However, after studying this piece of professional comment and a similar piece of comment by the other reviewer, we have realized that we should have provided these

data in our initial submission. Now, we have added all these data to the revised manuscript accordingly (see section 2.1-2.2, Fig.1, Fig2 for data of health status indexes before microbiota transplantation. Line100-123) (see section 2.5 and Fig.4 for analysis of microbial community composition between three groups of recipient mice before microbiota transplantation. Line141-151) (see section 2.6 and Fig.5 for analysis of microbial community composition before and after microbiota transplantation. Line152-164). The corresponding method information has also added to the revised version (Line421-423, Line427-433 and Line435-437 for data of health status indexes before microbiota transplantation) (Line447 for analysis of microbial community composition before microbiota transplantation).

Question-6

Line 319: Using only one aliquot to determine the total bacteria number does not leave room for error. Should this not have been done in biological or at least technical replicates?

Response-6

Thanks for the reviewer's comment. The following two paragraphs in blue color are the detailed procedure by which the total bacteria number in grafts samples were determined:

The substance in the small intestine of all donor mice were mixed and homogenized before being divided into 140 aliquots with the same volume (0.2 mL). The same procedure was performed with the substance in the large intestine of all donor mice. Therefore, the graft samples received by each of the recipient animals within a group were of the same microbial composition. One aliquot (0.2 mL) was used for quantifying the total bacteria number in a single aliquot microscopically using a Petroff-Hauser counting chamber.

Before counting the bacteria in the small intestine microbiota graft sample, one aliquot (0.2 mL) sample was first diluted in 1:5 and then was divided into five equal aliquots (0.2 mL/aliquot). Each of the five equal aliquots was repeatedly used for counting using a Petroff-Hauser counting chamber. And the mean number was calculated from the five raw counting values. Before counting the bacteria in the large intestine microbiota graft sample, one aliquot (0.2 mL) sample was first diluted in 1:500 and then was divided into 100 equal aliquots (1 mL/aliquot). After trying to counting using one of the 100 aliquots, we found there were still too high concentration of bacteria in it to counting using a Petroff-Hauser counting chamber. So, we randomly selected 5 of the rest 99 aliquots (1 mL/aliquot) and further diluted them again in 1:100 before subjected to counting. Similarly, the mean number was calculated from the five raw counting values.

The purpose of counting bacteria number was mere to know the numbers of bacteria contained in both grafts. The counting results were yet not used to in the process of the determination of how much bacteria should be transplanted for

each recipient mouse. Because by the method for grafts preparation and transplantation in our study, the total number of the small intestinal microbiota received by one recipient mouse throughout the whole transplantation was actually about the number of the small intestinal microbiota that one donor mouse provided. The same was for the large intestinal microbiota.

Besides, the samples were mixed enough by pipetting repeatedly so as to ensure a minimized error in microbiota number among each of the aliquots. Moreover, the principle of randomization was strictly obeyed when each of the aliquots was assigned to the recipient mice. **Therefore, the homogeneity of the number and composition of bacteria in the aliquoted samples of the same graft received by each recipient mouse were not affected by the possible errors in the process of counting and estimating the number of bacteria.**

Nevertheless, there were actually five technical replicates that minimized the error deriving during following counting process, although the aliquots divided from the initially prepared grafts samples after the first division were not repeatedly used for the following counting procedure. Now, we have added the required information about counting bacteria number to the revised manuscript (Line398-410)

Question-7

Line 197-199: Could these changes observed with SIMT be the result of a lack of diversity between the small and large intestine? Perhaps in SIMT, engraftment occurred at the small intestine because the composition of SIMG was similar to the small intestinal environment, whereas LIMT engraftment occurred only in the large intestine

Response-7

Thank the reviewer for his/her professional comment. As far as we understand, the reviewer means here that in this study, intragastric transplantation was used, in which the graft first entered the small intestine before entered the large intestine with downstream of the contents in small intestine. Because the environment in small intestine was more suitable for bacterial colonization of small intestine microbiota graft, significant effects were observed in the small intestine microbiota transplantation group. In contrast, bacteria in large intestinal microbiota graft were not easy to survive or thrive in the small intestine and therefore failed to exert significant effects.

This possibility may explain the different effects of small intestine microbiota transplantation and large intestine microbiota transplantation in this study. This point had not been included in the discussion of our initial manuscript. We are grateful that the reviewer pointed out and we have improved the discussion in the revised manuscript accordingly (Line311-318).

In fact, we had taken this problem into account and carried out a preliminary experiment at the very beginning of this study. The preliminary experiment told us there were insurmountable difficulties. The following **three** paragraphs are the detail information:

To explore whether possible different effects induced by small intestine microbiota transplantation and large intestine microbiota transplantation is due to '*a lack or a presence of diversity between the small and large intestine*' as the reviewer pointed out, our preliminary experiment employed such two designs: injecting the small intestine microbiota graft into small intestine and injecting the large intestine microbiota graft into large intestine; injecting the small intestine microbiota graft into large intestine and injecting the large intestine microbiota graft into small intestine.

However, for intra-small or intra-large intestine injection, the abdominal wall must be surgically opened, and the upper jejunum and cecum/appendix were found out before the bacteria-containing graft suspension was injected with a fine needle. As we tried, this transplant method brought to all mice, including the control mice, with a series of serious postsurgical sick-like implications such as drowsiness, decreased food intake, weight loss, elevated body temperature, constipation, reduced social interactions than before surgery. This suggested that the stress by abdominal wall incision might constitute severe surgical stress. Moreover, a possible leakage of intestinal bacteria into the peritoneal cavity after intestinal puncture might also occurred that would result in peritonitis. These events were bound to put the recipient mice in a state of disease. Therefore, the animals received this surgical procedure ought to be esteemed as more an animal model of postoperative stress after abdominal incision combined with enterocentesis than a normal/healthy mice model. The later, however, is indeed that we aimed to use for the evaluation of effects by **physiological** gut microbiota transplantation.

In addition, straightly injecting bacterial graft into large intestine is not proper because the transplanted bacteria would not be exposed to the stomach and small intestine. Their exposure to the small intestinal environment may be important in their colonization and interaction with the host because all naturally happened exposure/colonization/interaction with the host of gut microbiota in the gastrointestinal tract possessing absorption function starts from the stomach and then the small intestine before the large intestine. Besides, although harboring much smaller number of bacteria than the large intestine, the small intestine is as about seven times in length as the large intestine and is of more abundant blood circulation, larger mucosal surface area and stronger absorption ability. In sum, straightly injecting bacterial graft into large intestine is not only consistent with the widely used intragastric administration method, can also not mimic the natural process of intake/colonization/interaction with the host of gut microbiota in real life of human beings and animals.

Based on the reasons above and according to the literatures that used fecal microbiota transplantation in rodents, we decided to make both the small intestine microbiota

transplantation and large intestine microbiota transplantation intragastrically. Given the most widely utility in clinical practice and animal experimental research in gut/fecal microbiota transplantation, using intragastric transplantation allows best us to compare the results of our study with previous reports, and ensures best that our findings could be referred to in the future basial and even clinical researches in this field.

Besides the possible '*lack or presence of diversity between the small and large intestine*' as the reviewer pointed out', other factors might also underlie the different effects induced by small and large intestine microbiota transplantation. For instance, the difference in the composition of small intestine bacterial graft and large intestine bacterial graft lies mainly in the relative abundance of different bacteria, because most high abundance bacteria exist both in the small intestine and large intestine merely with different relative abundance values. Therefore, the bacteria, at least the common bacteria, in both grafts will share the opportunity to survive and reproduce in the small intestinal environment. In this line, so is in the large intestinal environment. What's more, the interaction between the grafted and host-holding microorganisms, the interaction between these microorganisms and intestinal mucosa, immune system, nutrition or metabolism are all very complex processes hard to be explored. These processes are very likely to participate in the different effects induced by small and large intestine microbiota transplantation in our study. In conclusion, it is indeed difficult to reveal exactly the mechanism underlying the different effects induced by small and large intestine microbiota transplantation.

Nevertheless, the data obtained in our study are sufficient to support our scientific hypothesis that small intestine microbiota transplantation and large intestine microbiota transplantation may exert different influences on their host.

We have added this contents to the discussion in the revised manuscript accordingly (Line318-340)

Minor Concerns:

Question-8

Figure 1: It is clear through plots on grafts A-D, that eating and drinking habits varied between the individual mice in each group. Why were these factors were not considered before transplantation given that the variations between individuals could expectedly impact the results.

Response-8

Thank the reviewer for his/her professional comment. As stated above, the health status indexes including food intake, water consumption, body weight, stool pellets number, stool wet weight and stool water percentage of both the donor mice and the

recipient mice before microbiota transplantation had also been recorded and analyzed with no significant differences found between three recipient groups or between the recipient and the donor mice. Therefore, we thought that it was unnecessary to report these data. Now, we have added all these data to the revised manuscript accordingly (see section 2.1-2.2, Fig.1, Fig2 for data of health status indexes before microbiota transplantation. Line100-123) (see section 2.5 and Fig.4 for analysis of microbial community composition between three groups of recipient mice before microbiota transplantation. Line141-151) (see section 2.6 and Fig.5 for analysis of microbial community composition before and after microbiota transplantation. Line152-164). The corresponding method information has also added to the revised version (Line421-423, Line427-433 and Line435-437 for data of health status indexes before microbiota transplantation) (Line447 for analysis of microbial community composition before microbiota transplantation).

As the reviewer pointed out, it is indeed that food intake and water intake varied between the individual mice in each group (Fig.1A-B in the initial submission or Fig.3A-B in the revised submission). Due to the smaller variation of food intake and water intake measures before starting microbiota transplantation (on PND75), the mechanism underlying the larger variation of food intake and water intake measures shown in Fig.1 in the initial submission involved likely the operation related to microbiota transplantation such as grabbing stress and intragastric action, because it emerged after the starting of microbiota transplantation. Although the reason for the larger variation in these indexes is not clear, this variations between individuals may be statistically distinguished by One-way ANOVA from the treatment-induced effects between groups. The later was what we aimed to observed.

Question-9

Line 78-79: Please provide a reference for the sentence "compared with microbiota in the small intestine, microbiota in the large intestine is more similar to fecal microbiota."

Response-9

Thanks for this piece of comment. We have now added two reference for this sentence: PMID:24116019 and PMID:25688558 to the introduction cited as 16-17. (Line78)

In the paper PMID:24116019, the author said that '*the results suggested the higher similarity of samples from large intestine and feces than small intestine and stomach*'.

In the paper PMID:25688558, the author said that '*the microbial profile in large intestine was more similar to feces than those in the small intestine*'.

Question-10

Line 304-325: Please clarify whether the donor microbiota was pooled before homogenization or if each donor was homogenized individually.

Response-10

Thanks for this piece of comment. As described in Line314 and Line 323-325 in the initial submission, the substance in the small or that in the large intestine of all donor mice were mixed and homogenized and then the homogenized substance was subjected to the succeeding steps. That is, the substance in the small intestine or that in the large intestine of all donor mice were mixed **before** homogenized. We have now made a more exact description about this information in the revised manuscript (Line386-387, Line396-398).

Question-11

Line 316: It is unclear if centrifugation was done performed on the particles removed in 312 or the filtered substance.

Response-11

Thank the reviewer for his/her professional comment. The particles were removed by passing through stainless steel laboratory sieves. No centrifugation was performed on the particles either before or after this removing. This method steps were determined according to the references cited in our manuscript: PMID: 28760163. Avoiding to many centrifugation/resuspension cycles will benefit the maintenance of normal vitality and relative abundance of the bacteria in the graft samples.

Question-12

Line 338-339: Why was the body weight recorded for the LIMT and SIMT group but not the control group? Your results state otherwise, please indicate that these measures were also done for your control group.

Response-12

Thank the reviewer for his/her careful comment. We are very sorry for the mistake in describing this information. We have now revised this statement accordingly (Line421-433).

Question-13

These results show that only change was seen in the large intestine given FMT, did the small intestine change? If the small intestine closely matched the composition of the LIMT than the idea would not be correct.

Response-13

Thank the reviewer for his/her comment. We are sorry that we cannot understand the exact meaning of this piece of comment. The abbreviation FMT usually stands for fecal microbiota transplantation. In our study, only small intestine microbiota transplantation (SIMT) and large intestine microbiota transplantation (LIMT) were carried out without FMT.

The aim of the present study was to investigate whether that small intestinal microbiota and large intestinal microbiota transplantation may exert different influences on their host. In brief, this study found that transplanting small intestinal flora significantly altered fecal flora, defecation, cytokines and mood status in healthy recipient mice, whereas transplanting large intestinal flora brought no significant influences.

Both small intestine microbiota transplantation and large intestine microbiota transplantation were performed intragastrically and only fecal samples, and not samples from small intestine or large intestine, were analyzed for the potential changes in the microbiota composition of the recipient mice.

These contents may contain the wanted information. If not, we will be grateful that the kindly reviewer gives us more comments or suggestions.

July 7, 2021

RE: Life Science Alliance Manuscript #LSA-2021-01039-TR

Dr. Junhua Yang
Guangdong Pharmaceutical University
Department of Anatomy, School of Biosciences & Biopharmaceutics
Room 510#, Building of Basic Medical Sciences, School of Biosciences & Biopharmaceutics,
Guangdong Pharmaceutical University, No.280, East Waihuan Street, Higher Education Mega
Center, Guangzhou Cit
Guangzhou, Guangdong 510006
China

Dear Dr. Yang,

Thank you for submitting your revised manuscript entitled "Small Intestinal Flora Graft Alters Fecal Flora, Stool, Cytokines and Mood Status in Healthy Mice". We would be happy to publish your paper in Life Science Alliance pending final revisions necessary to meet our formatting guidelines.

- please add a Summary Blurb/Alternate Abstract in our system
- please make sure the author order in your manuscript and our system match
- please consult our manuscript preparation guidelines <https://www.life-science-alliance.org/manuscript-prep> and make sure your manuscript sections are in the correct order
- please leave only the final version of your manuscript uploaded
- please add callouts for Figure 2A-F to your main manuscript text
- please incorporate the Supplemental Materials into the main Materials & Methods section

LSA now encourages authors to provide a 30-60 second video where the study is briefly explained. We will use these videos on social media to promote the published paper and the presenting author. Corresponding or first-authors are welcome to submit the video. Please submit only one video per manuscript. The video can be emailed to contact@life-science-alliance.org

A. FINAL FILES:

B. MANUSCRIPT ORGANIZATION AND FORMATTING:

Sincerely,

Eric Sawey, PhD
Executive Editor

Reviewer #1 (Comments to the Authors (Required)):

The authors have provided a detailed response to the several of the queries raised. No further comments.

July 14, 2021

RE: Life Science Alliance Manuscript #LSA-2021-01039-TRR

Dr. Junhua Yang
Guangdong Pharmaceutical University
Department of Anatomy, School of Biosciences & Biopharmaceutics
Room 510#, Building of Basic Medical Sciences, School of Biosciences & Biopharmaceutics,
Guangdong Pharmaceutical University, No.280, East Waihuan Street, Higher Education Mega
Center, Guangzhou Cit
Guangzhou, Guangdong 510006
China

Dear Dr. Yang,

Thank you for submitting your Research Article entitled "Small Intestinal Flora Graft Alters Fecal Flora, Stool, Cytokines and Mood Status in Healthy Mice". It is a pleasure to let you know that your manuscript is now accepted for publication in Life Science Alliance. Congratulations on this interesting work.

*****IMPORTANT:** If you will be unreachable at any time, please provide us with the email address of an alternate author. Failure to respond to routine queries may lead to unavoidable delays in publication.*******

DISTRIBUTION OF MATERIALS:

Again, congratulations on a very nice paper. I hope you found the review process to be constructive

and are pleased with how the manuscript was handled editorially. We look forward to future exciting submissions from your lab.

Sincerely,
